# Polymeric Materials, Advances and Applications in Tissue Engineering: A Review

**DOI:** 10.3390/bioengineering10020218

**Published:** 2023-02-06

**Authors:** María Cecilia Socci, Gabriela Rodríguez, Emilia Oliva, Shigeko Fushimi, Kiyofumi Takabatake, Hitoshi Nagatsuka, Carmelo José Felice, Andrea Paola Rodríguez

**Affiliations:** 1Laboratorio de Medios e Interfases (LAMEIN), Departamento de Bioingeniería, FACET-UNT, Tucumán 4000, Argentina; 2Instituto Superior de Investigaciones Biológicas (INSIBIO), CONICET, Tucumán 4000, Argentina; 3Department of Oral Pathology and Medicine, Faculty of Medicine, Dentistry and Pharmaceutical Sciences, Okayama University, Okayama 700-8525, Japan; 4Department of Oral Pathology and Medicine, Okayama University Dental School, Okayama 700-8525, Japan

**Keywords:** tissue engineering, biomaterials, scaffolds, regenerative medicine, stem cells

## Abstract

Tissue Engineering (TE) is an interdisciplinary field that encompasses materials science in combination with biological and engineering sciences. In recent years, an increase in the demand for therapeutic strategies for improving quality of life has necessitated innovative approaches to designing intelligent biomaterials aimed at the regeneration of tissues and organs. Polymeric porous scaffolds play a critical role in TE strategies for providing a favorable environment for tissue restoration and establishing the interaction of the biomaterial with cells and inducing substances. This article reviewed the various polymeric scaffold materials and their production techniques, as well as the basic elements and principles of TE. Several interesting strategies in eight main TE application areas of epithelial, bone, uterine, vascular, nerve, cartilaginous, cardiac, and urinary tissue were included with the aim of learning about current approaches in TE. Different polymer-based medical devices approved for use in clinical trials and a wide variety of polymeric biomaterials are currently available as commercial products. However, there still are obstacles that limit the clinical translation of TE implants for use wide in humans, and much research work is still needed in the field of regenerative medicine.

## 1. Introduction

Human beings are made up of multiple complex tissues assembled in hierarchical structures that range from macro to nanoscale and fulfill specific roles to maintain the proper functioning of the body. These biological characteristics and structures inspired many scientists to design advanced multifunctional materials for the replacement of organs and tissues [1]. Every year, millions of patients suffer total or partial damage to their organs and tissues, and they are potential candidates for studies in the field of regenerative medicine. On the other hand, human life expectancy has quadrupled in the last three centuries, and the great drawback of conventional treatments lies mainly in the difficulty of finding donors and the rejection of the transplanted organ/tissue by the recipient organism [2,3]. This is how the field of TE in biomedicine was born, in order to develop functional tissues capable of regenerating and/or improving damaged tissue, and do so requires the contribution of several fields: TE, cell therapy, molecular therapy (e.g., gene and drug delivery), and artificial and bio-organ technology. TE started as a branch of regenerative medicine, and it is a rapidly growing research field in recent times. It combines engineering and biological science principles to create functional substitutes for native tissue and facilitate the maintenance, repair, and restoration of damaged tissue. In recent years, it has received considerable attention, as it is a promising field with a likely profound impact in field of medicine [4,5].

The term TE was officially created at a National Science Foundation workshop in 1988. TE consists of “the application of engineering and life science principles and methods toward the fundamental understanding of the structure-function relationship in normal and pathological mammalian tissues and the development of biological substitutes to restore, maintain, or improve tissue function” [6]. The roots of TE as a modern discipline lie in Boston, and the first recorded use of the term TE, as applied today, was published in an article titled “Functional Organ Replacement: The New Tissue Engineering Technology” in “Surgical Technology International” in 1991 [7]. Although the field of TE appears to be relatively new, the idea of replacing one tissue with another is as old as history itself. Examples are the Greek legend of “Prometheus” and the eternal regeneration of his liver, the miracle of the creation of Eve in “Genesis”, and the miraculous transplant of a member of the Holy Cosmos and Damian. With the introduction of the scientific method and advancements in our knowledge of traumatic injuries and diseases, the secrets of biology are better understood now [8].

TE and regenerative medicine are interdisciplinary fields that have evolved rapidly in recent years (Figure 1), and TE focuses on repairing and restoring the structural function of damaged tissue using various materials such as decellularized matrices, cells, scaffolds, and others [9]. A scaffold is a three-dimensional platform that can mimic the extracellular matrix (ECM) and is capable of providing mechanical, spatial, and biological signals to regulate and guide cellular responses [10] (Figure 2). Many researchers [11,12,13,14,15,16] use polymeric matrices composed of a mixture of one or more polymers (natural or synthetic) to solve the mechanical, thermal, and biological challenges that arise when they are used as scaffolds. Currently, the focus is on developing smart material devices that combine the benefits of different components, taking into account the specific biological, clinical, and medical aspects of the tissue defect.

In regenerative medicine, cell management strategies can help replace lost cells in cases where the endogenous cells are insufficient or dysfunctional (e.g., regeneration of nerve tissue). TE tries to combine aspects of biomaterial scaffolds with cell replacement techniques to create a 3D environment that influences cell behavior (of native cells and cells previously cultured in the implant), such as their phenotype, architecture, migration, and survival. Biomaterials can provide structures for host cell infiltration, differentiation, and organization, and can serve as a means for drug delivery (e.g., controlled release of nanomedicines) [17]. These composites (scaffold-cells) combined with suitable biomolecules of interest (cytokines, growth factors, adhesion proteins, peptides, etc.), can be cultured in vitro to prepare a tissue that will subsequently be implanted in the damaged area and regenerate tissue in vivo. This combination of cells, signal molecules, and scaffolds is known as the tissue engineering triad [6] (Figure 3).

In this review, we highlight the importance of the use of polymers in ET, followed by brief descriptions of different methodologies for their fabrication. In addition, we put emphasis on the description of polymeric biomaterials, since they are mainly chosen for the design and construction of scaffolds due to their wide range of properties, availability, and low cost. Moreover, we provide a concise review of recent advances in the preclinical applications for different tissue types (epithelial, skeletal, cartilage, urinary, uterine, nervous, cardiac, and adipose). Finally, we gather some relevant data for the application of polymeric matrices in clinical trials, and we make a brief description of the current state of the polymers available as commercial products.

**Figure 1 bioengineering-10-00218-f001:**
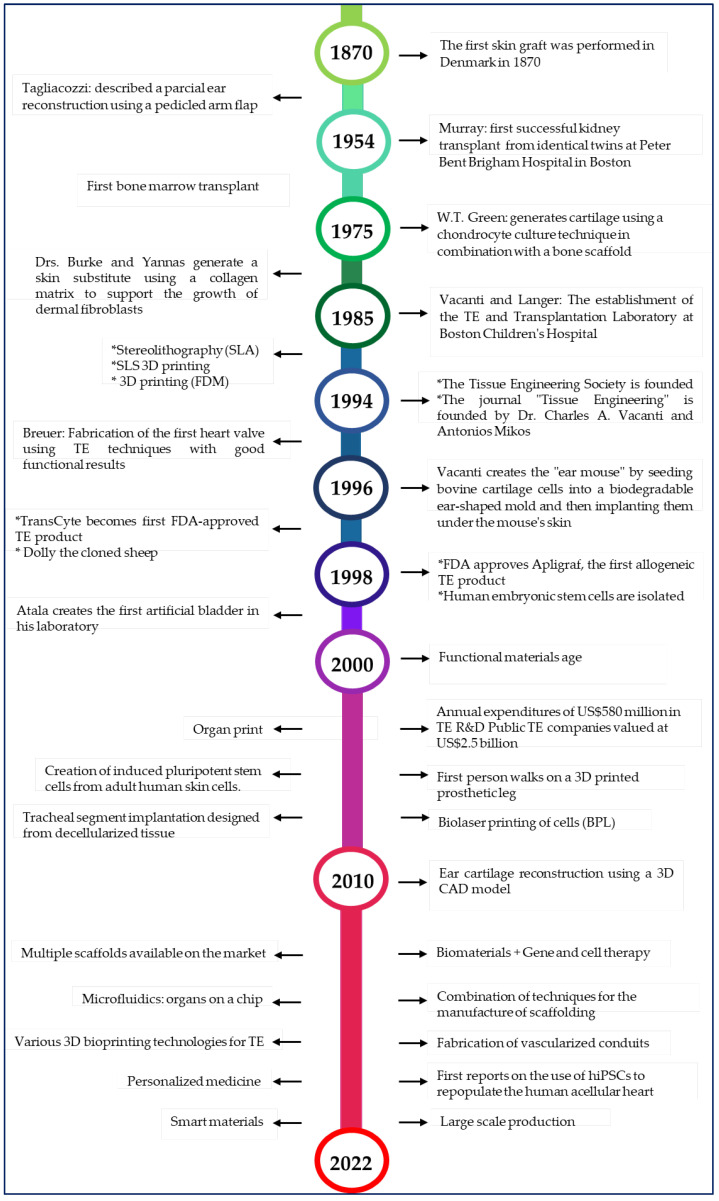
Timeline of tissue engineering [6,7,8,18,19,20,21,22,23,24,25,26].

**Figure 2 bioengineering-10-00218-f002:**
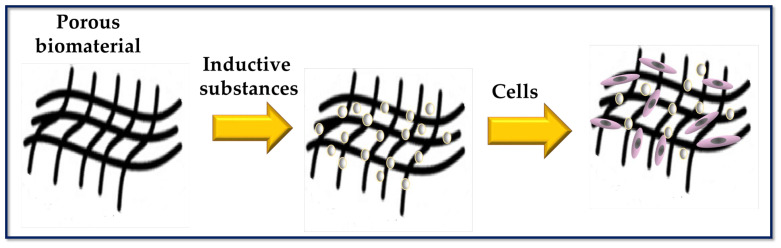
Scheme to represent a porous biomaterial used as a scaffold for tissue engineering. The surface of the scaffold is functionalized with biomolecules capable of inducing cellular responses.

**Figure 3 bioengineering-10-00218-f003:**
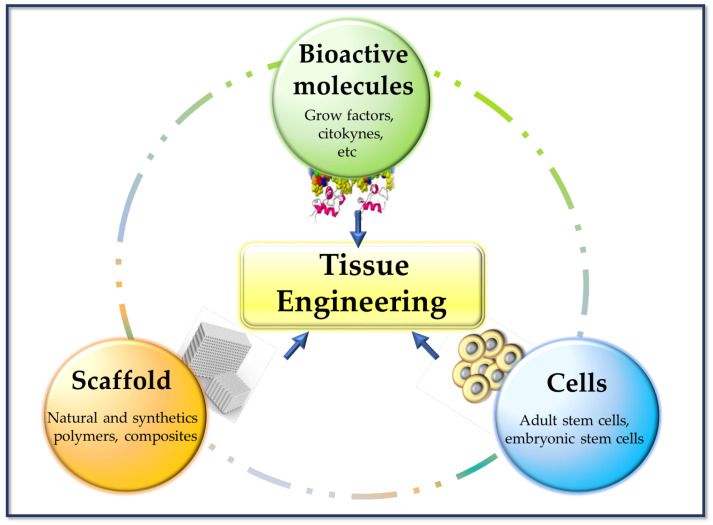
Basic pillars of tissue engineering.

## 2. Biomaterials for Tissue Engineering Scaffolds Fabrication

The definition of biomaterial emerged in 1976 at the First Consensus Conference of the European Society for Biomaterials (ESB). A biomaterial was defined as “a non-viable material used as a medical device, intended to interact with biological systems”. However, the current definition of biomaterial by ESB is “material intended to interact with biological systems to evaluate, treat, augment, or replace any tissue, organ, or function in the body” [6]. This subtle change in definition is indicative of how ESB has evolved in the field of biomaterials over time. 

The repair and healing of the tissue usually involves the autograft technique as a conventional method, but it depends mainly on the availability of other donor tissue and on factors such as graft failure, pain, persistent bleeding, the risk of associated infectious diseases in the patients, etc. [27,28]. Current TE technologies have an advantage over traditional technology, since studies have focused on choosing materials with beneficial characteristics to build scaffolds that are highly compatible with the human body and its tissues. This is of great importance as it reduces the risk of immune rejection and the susceptibility to infection [29]. Biomaterials can be used as implants in the form of sutures, bone, joint replacements, ligaments, vascular grafts, heart valves, intraocular lenses, dental implants, and medical devices such as pacemakers, biosensors, etc. [30]. Therefore, bio-based materials are changing the dynamics of 21st century materials. The use of bio-based materials in various research areas is becoming more frequent, as they have potential applications in health care to improve the quality of life of many people [31].

An important concept in TE strategies is that of the scaffold, defined as a highly porous three-dimensional (3D) matrix capable of providing an adequate surface for cells adhesion and interacting with the biomolecules of interest (Figure 4). The composition and internal architecture of the scaffolds control cell behavior and well-being [6,32] by acting as a supporting prosthesis in vivo or as a cell adhesion substrate for TE in vitro [33]. The scaffolds shape the macroscopic level of the organs and tissues to be replaced without recreating the details that are observed at the nanoscale in real organs (Figure 5). However, the nanoarchitecture of the ECM provides an intricate fibrillar system in which specific molecular interactions occur between various ratios, isoforms, and geometric shapes of elastins, collagens, proteoglycans, and adhesion proteins such as laminins and fibronectins. This creates an environment with informational signals and instructions that guide cellular behavior to form complex tissues such as bone, liver, heart, and kidney tissue [34].

Currently, various approaches can be applied to improve the biological and mechanical characteristics of scaffolds by combining design and manufacturing strategies with new materials. For the design of a good scaffold, it is of great importance to evaluate certain criteria before choosing a technique or its manufacture (Table 1). Parameters such as the surface characteristics (topography and roughness) and porosity (pore shape and size) of the scaffold must be controlled to increase cell migration into and on the surface of the scaffold [36,37] and to favor the efficient transport of metabolites without significantly compromising its mechanical stability [38]. The final applications of the scaffolds can vary significantly. They can be implanted empty (acellular) when they are expected to be colonized and invaded by host cells in a short time, or they may need to be previously seeded with appropriate cells before implantation or pre-cultured with cells in vitro in a suitable culture medium [39]. It is important to note that most mammalian cells depend on anchorage. Cell adhesion plays a crucial role in the development and maintenance of tissues, since it stimulates the signals that regulate the cell cycle, migration, differentiation, and cell survival. Therefore, it is important to design scaffold matrices that increase cell affinity to promote an ideal environment for cell attachment [40,41].

An ideal TE biomaterial not only mimics the native ECM of the tissue and provides mechanical support but also allows vascularization and integration with the host tissue, and gradually biodegrades and is remodeled with time as new tissues are formed. In this way, the native tissue can integrate with the scaffold and gradually replace the area originally occupied by it [42,43].

**Table 1 bioengineering-10-00218-t001:** Criteria for building an ideal scaffold.

Feature	Description
Adequate intrinsic physical and mechanical properties	This is defined by the microarchitecture and surface microtextures (surface topography). An ideal microarchitecture should be highly porous, with defined and interconnected pore sizes and a high surface area to volume ratio to allow for better vascularization, mass transfer, and cell growth.
Biocompatibility	It must produce the desired effect, be safe, and cause the minimum degree of inflammation once implanted.
Bioactivity	The biomaterial-cell interaction favors cell adhesion and proliferation, facilitating contact between cells and their migration over a prolonged period. Therefore, scaffolds can include biological molecules on their surface to promote cell adhesion or can also serve as a delivery vehicle or reservoir for growth-stimulating substances such as growth factors to accelerate regeneration.
Mimic EMC	It must be capable of mimicking the native tissue, providing an environment of optimal protection and nutrition.
Bioabsorption	It must be bioabsorbed in a controlled and appropriate time so that the new tissue replaces the space initially occupied by the biomaterials.
Versatility	They must be adaptable to different manufacturing techniques.
Translational perspective	The scaffold must be reproducible, accessible, and scalable to enable its use in high-demand applications for large tissues.

Bibliography consulted [4,30,39,44].

## 3. Polymers for Tissue Engineering

When the tissue is damaged after an accident or illness, it is not always possible to fully recover or have access to a donor in a short time. This is where polymer-based scaffolds can play a vital role in improving the quality of life of the patient by restoring tissue and maintaining a suitable environment to produce faster healing [29]. Polymer engineering represents a growing area that is often tailored to specific needs in terms of the design and manufacturing for a given application [45]. Although there are various materials available to produce scaffolds (Figure 6), they are usually chosen for the design and construction of scaffolds due to their wide range of properties, availability, and low cost [46]. In addition, they stand out as a special class of materials due to their flexibility and versatility. They have been used for many types of biomedical devices, such as dental [47], orthopedic [48], and cardiovascular [49] implants.

A polymer is a long-chain macromolecule composed of repeating subunits called monomers that are connected by covalent bonds, and knowledge of the synthesis and manufacture of products from them is essential in the biomedical field [50]. Among the various classes of biomaterials available for medical uses (Figure 6), natural polymers (NP) and synthetic polymers (SP) are used the most often for the manufacture of scaffolds.

NPs are produced by biological systems and have been used in human applications such as pharmaceutical excipients, drug delivery, cosmetics, prostheses, and biomedical scaffolds [51]. NPs are components of the ECM, and these bioactive properties allow them to be recognized and degraded by the biological environment, increasing cell interaction with the biomaterial [52]. Also, they present less toxicity from chronic inflammation or immunological reactions than those observed with the use of SP [53] and can undergo chemical modifications and be potentially biodegradable and biocompatible [54]. For all of these reasons, as well as for their cost effectiveness and ready availability, the use of NPs is attractive in biomedical applications. Their disadvantages are their sensitivity to temperature increase, which causes these polymers to be destroyed before reaching their melting point; their complex structure, which makes them difficult to process; and the possibility of the transmission of diseases to humans from other species due to their sources of origin [52]. There are several examples of NPs that are used in clinical applications. Among the proteins are silk fibroin, collagen, gelatin, albumin, keratin, fibrinogen, elastin, and actin; the polysaccharides include chitosan, chitin, alginate, gellan gum, and derivatives; and the glycosaminoglycans include hyaluronic acid [30,53].

SPs appeared much later than NPs but have been playing crucial roles in recent times. They are more diverse and versatile for biomedical applications due to the ease of creating custom designs with them and making more controlled chemical modifications [53]. They also offer several further advantages over other materials used in developing TE scaffolds, such as the ability to tailor their mechanical properties and their more controlled degradation kinetics for various applications [55]. Often cheaper than biological scaffolds, they can be produced uniformly in large quantities and have a long storage time. Many commercially available SPs exhibit physical, chemical, and mechanical properties comparable to those of biological tissues [51]. The most widely used SPs approved by the FDA (Food and Drug Administration of the United States) are aliphatic, e.g., poly (lactic acid) (PLA), poly (glycolic acid) (PGA), and its copolymers poly (lactic-co-glycolic acid) (PLGA). They are linear aliphatic polyesters and degrade by the hydrolysis of their ester linkages, resulting in non-toxic products and metabolites that can be removed by the natural metabolism of the host [56]. Other linear aliphatic polyesters, such as poly (ε-caprolactone) (PCL) and poly (hydroxybutyrate) (PHB), are also used in TE research. PCL degrades at a significantly slower rate than PLA, PGA, and PLGA, making PCL less attractive for general TE applications, but more attractive for long-term implantation and the controlled release of substances [57]. Other SPs widely applied in TE to manufacture scaffolds are poly (propylene fumarate) (PPF) and poly (glycerol sebacate) (PGS). PPF is a good candidate for bone TE application as it is biocompatible, biodegradable, and osteoconductive and can be strengthened by cross-linking reactions [56]. Polymers are generally degraded by hydrolysis, producing intermediate natural metabolites, and have controllable degradation rates ranging from months to years. Therefore, scaffolds could be designed according to the requirement of each tissue by adjusting the initial proportion of the monomers. In addition, this is an advantage when polymers are required to release a molecule of interest, such as hormones and growth factors, in a controlled manner [19]. Despite all these advantages, many SPs tend to present an immune response or toxicity when combined with certain polymers that the host tissue cannot incorporate [53]. Therefore, one possible solution is to combine synthetic and natural polymers to overcome the deficiencies of each and obtain a scaffold that exploits the great merits of both [58].

## 4. Strategies for Fabrication of Scaffolds in Tissue Engineering

To obtain biomedical devices with improved diagnostic and therapeutic capabilities, there are modern technologies in their design and manufacturing methods that promote remarkable alternative approaches in the field of regenerative medicine and TE. These include new micro- and nano-fabrication technologies, tools based on the use of medical images, computer-aided design and manufacturing, rapid prototyping and manufacturing technologies, and current advances in materials science [59]. Some of the most widely used techniques for scaffold fabrication in TE are described below.

### 4.1. Electrospinning

It is a unique, simple, cost-effective, and efficient technique for manufacturing continuous polymeric, ceramic and hybrid fibers. The diameters are in the range of hundreds of nanometers. Although it has many other potential applications, electrospinning has taken on great importance in TE, since it can be applied in the production of large-scale nanostructured scaffolds for tissue growth [60,61]. As for the equipment, the main components include a power supply of high voltage (can be direct or alternating current), an infusion pump coupled to a syringe, a capillary (usually a blunt-tipped hypodermic needle), and a conductive manifold [62] (Figure 7). During the electrospinning process, a high voltage is applied to the polymeric fluid (several tens of Kv), which is delivered to the tip of the capillary. This causes an accumulation of charges on the free surface of the fluid that protrudes from the tip of the needle and interacts with the external electric field. When the same polarity charge is reached as in the solution, a repulsive Coulomb force is generated which, when it overcomes the surface tension of the drop, allows the formation of a Taylor cone. From that, a continuous fluid jet is expelled which is elongated and accelerated by the applied electric field, travels a short distance between the electrodes, and is finally deposited on the collector of the opposite charge. As a result, the process leads to the formation of fine solid fibers when the solvent evaporates [63,64,65,66]. Random fiber ordering is generally achieved, but aligned nanofibers can also be obtained using a fast-rotating collector or a pair of parallel electrodes. This arrangement is useful in nerve or skeletal muscle TE applications [67]. Moreover, there is coaxial electrospinning that allows the combination of solutions with different properties into a single fiber, and it can be useful for the encapsulation and protection of functional molecules and their controlled release [68].

An important factor to control in electrospinning is the diameter of the fibers, which depends on the processing parameters and conditions (injection flow, applied voltage, needle-collector distance, etc.), the properties of the polymer solution (viscosity, conductivity, polymer type, concentration, surface tension, etc.), and the environmental conditions (relative humidity, temperature, and atmospheric pressure) [69,70]. All of these can dramatically influence the resulting diameter, shape, and spatial distribution of the electrospun fibers [71]. Various benefits associated with electrospun fibrous structures are their nanometric structure that mimic the architecture of the ECM, their large surface/volume ratio for binding cells and bioactive factors, their highly interconnected porous architecture, their high load capacity, their encapsulation efficiency, and their capability of transporting various drugs to specific sites of action [72,73,74].

### 4.2. Molecular Self-Assembly

Self-assembly describes the autonomous organization of individual components into ordered patterns or structures. It is characterized by the specific association of molecules through non-covalent interactions, including hydrogen bonding and ionic bonding, and hydrophobic and Van der Waals interactions. Individually, such interactions are weak, but, in large numbers, they dominate the structural and conformational behavior of the assembly [75]. The concept of self-assembly has been applied to the construction of synthetic nanostructures for TE that aim to mimic the ECM [76]. The building blocks of construction interact one by one at the atomic and molecular scale, to organize into more complex supramolecular structures that can further reorganize into functional native tissue structures [77]. Thus, a further understanding of the biomolecular process could inspire the assembly of nanoparticles to promote the design and manufacture of new functional nanomaterials [78] with personalized morphologies from a single molecule. To obtain a final product with the desired properties, it is necessary to control the modulation of the individual monomeric building blocks [79] and other parameters (self-assembly time, temperature, solution concentration, pH, metal ions, etc.) [80]. Highly ordered biomaterials can be obtained from zero-dimensional (0D) structures such as the nanoparticles, nanospheres, and rings that are base structures for protein formation; one-dimensional (1D) structures such as nanofibers or nanotubes; two-dimensional (2D) structures such as nanofilms or 2D DNA arrays; and three-dimensional (3D) structures such as 3D hydrogels [81]. Figure 8 shows an example of the different dimensions that materials can take on nanocarbon as modifiers to enhance the visible light photocatalytic activity of g-C3N4 [82]. The two types of natural materials usually used in manufacturing processes are collagen and elastin. Both molecules are found abundantly in nature because they are components of all connective tissues and the ECM. These molecules have been used as the basis for the design of new materials that resemble collagen and elastin, though they gave way to the de novo design of synthetic peptides. Currently, peptide-based hydrogels are used to provide extracellular environments that mimic ECM and open up great opportunities for biomedical applications in fields such as TE, drug delivery, and wound healing [83].

### 4.3. Phase Separation

In this process, the polymer dissolves and phase separation is induced, either thermally or by addition of an immiscible solvent to the polymer solution, leading to the formation of a gel [84,85]. The term TIPS refers to temperature-induced phase separation, and, in this process, polymer precipitation is caused by a decrease in temperature, normally provoked by immersion in a quenching bath. It is a conventional technique for manufacturing highly porous scaffolds by forming networks of interconnected pores [86,87]. It has the advantage of being a fast, inexpensive, scalable, and controllable procedure [88,89], but its application is limited to certain polymers compatible with the use of organic solvents [28]. The TIPS process can be divided into solid-liquid phase separation (SLPS) (Figure 9) and liquid-liquid phase separation (LLPS), depending on the crystallization temperature of the solvent [90]. In the first case, a homogeneous solution of polymer is exposed to a certain temperature and becomes thermodynamically unstable, resulting in two separate phases. The solution is then freeze-dried, and the polymer-rich phase forms a matrix, while the lean phase produces pores when the solvents are removed. The SLPS method involves the production of porous scaffolds with good cell interconnectivity and permeability [91]. In the second case, a non-solvent is added to produce two different morphologies: (1) the solution separates into two phases (one rich and one lean in polymer) giving an emulsion when cooled below the binodal curve, and (2) the solution separates into a bicontinuous polymer-rich phase and a lean phase which, when cooled below the spinodal curve, produces a polymer with a highly porous structure, once the solvent is removed by leaching or cold drying [87,92]. Theoretically, the polymer-rich phase consists of the polymer and a fraction of the solvent, and the polymer-lean phase contains a non-solvent and the remaining solvent [93]. It is important to control certain parameters, such as the temperature and rate of cooling, crystallinity of the polymer, polymer concentration, and presence of ceramic powders, to manipulate and control the pore size, the pore geometry, and the interconnectivity of the pores in the scaffold [94].

### 4.4. Particulate Leaching

Particulate leaching consists of two categories: (1) solvent casting—particulate leaching (SC-PL) and (2) melt molding—particulate leaching (MM-PL). In SC-PL, a polymer solution is mixed with salt particles in a mold; then, the solvent is evaporated and the salt particles are removed by washing with water, resulting in the formation of a porous scaffold. In the MM-PL, the polymer is formed by introducing it into a mold with the embedded solid porogen (Figure 10). Pressure and heat are then applied to evaporate the solvent, and the particles are leached by washing the resulting product with water [95,96]. The SC-PL method is relatively simple and does not require specific and expensive equipment; it is also possible to control the final porosity, the pore size, and the interconnectivity of the pores by making a suitable selection of the initial proportions of the polymer and the porogen [97]. One of the advantages is that highly porous scaffolds (some reaching 93%) can be obtained with average pore sizes of up to 500 µm. However, the distribution of salt particles may be non-uniform inside due to the density difference between liquid polymer and solid salt. The number of salt particles in direct contact with each other is not well controlled, and salt residues may remain despite washings, which will lead to the poor interconnectivity of the pores [98].

### 4.5. Gas Foaming

In this process, gases or blowing agents are mixed with polymeric solutions to manufacture porous matrices. Gas foaming can be carried out using chemical or physical methods. In the first case, the gases are formed from a chemical reaction or the decomposition (Figure 11) of a polymer salt complex in hot water, while, in the physical method, gases inert to the polymer are directly injected into the solution. The physical blowing agents are often volatile liquids that evaporate and cause the foam to expand, or pressurized gases or air injected directly into the foaming medium [99]. The polymers go through a pressurization process with gases such as N_2_ or CO_2_ over an extended period, until the full saturation level is reached. During the dissolution process, a monophasic homogeneous solution is formed. In the second stage, the nucleation process occurs due to thermodynamic instability, achieved by changes in temperature or pressure, leading to a decrease in the solubility of the gas, which escapes and forms bubbles within the polymeric matrix. Finally, the solution is stabilized by cooling [95,100]. Thus, it is possible to obtain pores that vary from 100 to 50 µm. When the gas formation is induced by a chemical reaction during the mixing process (e.g., N_2_), foam formation is rapidly induced, leading to the formation of a highly porous network. Although it is a relatively easy technique that does not require the use of solvents, most scaffolds made by this process have poorly connected pores and a non-porous outer surface, making them less suitable for TE applications [81].

Tang and collaborators proposed a new simple and cost-effective gas-foam method for fabricating hydrogels with a macroporous structure for application in bone defects in which they mixed a cell-laden hydrogel with Mg particles. The production of H_2_ gas occurred in situ during their degradation, generating a hydrogel with an optimized porosity and greater bioactivity [101].

### 4.6. Rapid Prototyping

Rapid prototyping (RP) is a group of dynamic and evolutionary technologies that create complex three-dimensional (3D) objects additively, layer by layer, from a predefined design file. The development of RP is closely related to the development of the computer and software industry. In particular, the development of computer-aided design (CAD) plays a critical role in the emergence of almost all RP systems today [32,102]. RP technologies play an increasingly important role in biomedical applications for TE, and they have also been applied in surgical planning, drug delivery devices, implants, and custom prosthetics. These systems are broadly classified into three categories based on the initial state of the material used, liquid, solid, and powder [103], and apply to polymers, ceramics, and metals [104].

Printing techniques for polymeric materials mainly include the Stereolithography Apparatus (SLA) [105], Fused Deposition Modeling (FDM) [106], 3D Printing [107], and Selective Laser Synthesis (SLS) [108]. The general scheme of RP is composed of (a) a computer with software to carry out the design, (b) machinery to carry out the additive process (or printers), and (c) suitable materials [109].

There is a variant within 3D printing, and it is an emerging technology with a great impact in the area of TE and regenerative medicine: 3D Bioprinting [110,111]. In conventional 3D printing for the fabrication of non-cellular scaffolds, strategies such as microextrusion, inkjet, and laser-assisted printing are used [112]. However, when we refer to bioprinting, biomaterials and cells are deposited simultaneously in an additive manner and can be considered as “bioinks” [113]. For a correct maturation of the tissue, the deposition of cells and biomaterials must be precise to facilitate cell-cell and cell-biomaterial interactions [114]. Due to advances in 3D printing technology and the increasing availability of user-friendly software, almost any shape can be designed, and this is highly attractive for the design and fabrication of custom surgical implants [115] (Figure 12 and Figure 13).

### 4.7. Decellularization

Decellularization is a technique that has been intensively studied in the last decade, and its objective is to produce artificial organs and tissues from the elimination of all their cells, while fully preserving the ultrastructure and composition of the native ECM [23] (Figure 14). With this aim, scaffolds of decellularized extracellular matrix (dECM) are achieved that serve as three-dimensional biological supports with vascular conduits that facilitate future grafting through vascular anastomosis. To produce these dECMs, cells can be removed from donor tissue using physical, chemical, enzymatic, or combined methods, resulting in the removal of the immunoreactive intracellular components [117,118,119]. These are obtained from various allogeneic or xenogeneic tissues, including the bladder matrix, dermis, heart valves, small intestinal submucosa, urinary bladder mucosa, skeletal muscle, mesothelium, or pericardium of different species using different methods [120].

Currently, numerous clinical products are composed of ECM and are used as surgical materials. These are obtained from various allogeneic or xenogeneic tissues, including the bladder matrix, dermis, heart valves, small intestine, mesothelium, or pericardium of different species [121]. As can be seen, the dECM is completely integrated with natural components, which gives it great advantages such as appropriate mechanical properties and biocompatibility [23]. In general, the host recognizes xenogeneic and allogeneic cellular antigens, and they are capable of inducing an inflammatory response, or the host’s immune system may reject the tissue. However, ECM components are generally conserved between species, so implantation can be tolerated [122]. The bioscaffolds (e.g., scaffold of biological origin) can be recellularized to create potentially functional constructs as a therapeutic strategy in regenerative medicine [120]. Although successful decellularization has been achieved for many organs, scientists and engineers are still making great efforts to design functional arrays and establish standardized decellularization protocols for clinical applications. To achieve those standardized protocols, it is essential to have solid knowledge of topics such as biodegradation, cytocompatibility, pathogenicity, and immunogenicity [123].

Interestingly, Aulino et al. mention a new concept, the multipotency of scaffolds, because the skeletal muscle acellular scaffolds offer a multipotent environment. These scaffolds were implanted at the interface of the tibialis anterior/tibial bone and the masseter muscle/mandible bone in a murine model and promoted muscle, bone, and cartilage formation at the interface with long and flat bone [124].

### 4.8. Cell Sheets

The effectiveness of tissue regeneration is based on a good integration of the scaffold with functional cells. However, despite extensive research efforts, some conventional applications of TE have not produced the desired results due to certain limitations such as a low cell survival rate, the inability of the injected cells to adhere to the injured site, the difficulty of vascularization, and an immunological rejection against the scaffold that may occur. To address this problem, several research groups have developed new TE approaches based on cell layer technology (CST). The CST is a novel technique that allows cells to be preserved in a sheet format without the need to use proteolytic enzymes or other destructive methods, enabling the preservation of ECM components [125]. These methods are promising for use in regenerative medicine because they preserve cell–ECM interactions, cell–cell contacts, and the composition of key proteins such as fibronectin, leukemia inhibitory factor receptor (LIFR), integrin-5, stromal cell-derived factor 1 (SDF-1), myosin heavy chain (MHC), endothelial growth factor (VEGF), and β-actin from cell membranes [126].

Sekine and Okano describe a method for creating tubular cardiac tissue for in vitro implanting and culturing cardiac cell sheets on the inner wall of a perfused segment of the small intestine [127], and Nishida et al. performed ocular surface reconstruction in four patients using autologous oral mucosal epithelial cells. They used a carrier-free cell sheet by exploiting the temperature-responsive culture surfaces by lowering the temperature; they thus managed to obtain an intact cell sheet to be transplanted without the need for a support [128].

With the CST system, numerous films of cells can be applied directly to a defective area as a coating creating a three-dimensional structure, or even shrunk into a cell pellet, which can be applied as a graft to the area of interest. [125]. Despite the great advantages offered by the CST technique, it is still difficult to obtain intact cell layers due to the low stability and low strength of the ECM. On the other hand, multilayer stratification can limit cells’ growth and nutrient supply, which reduces cells’ viability and differentiation potential. Therefore, new strategies combining CST systems with porous membrane supports are proposed [129]. In conclusion, CTS is a promising approach that should prove useful as a fundamental and widely used technique in next-generation ET and regenerative medicine [130].

## 5. Smart Materials

Materials that are described as “smart” or “functional” are usually part of a “smart system” that can sense and respond to its environment. They are designed to have one or more properties that can be significantly changed in a controlled fashion by external stimuli. The stimuli can be thermal, electrical, magnetic, mechanical, chemical, etc. Smart materials are the basis of many applications, such as textiles [131], food and beverage packaging [132], and biomaterials for tissue engineering [133], among others. In the present article, the focus is the use smart polymers to develop porous structures for tissue engineering. Thus, Mountaki et al. reported the synthesis of “smart”, pH-responsive, biodegradable polyesters bearing alkene/carboxylic acid side groups. The carboxylic acid side groups provide pH-responsive properties to the polymers and render them water soluble, whereas the alkene groups can be utilized for the formation of stable hydrogels and cross-linked polymer films [133].

One of the conventional methods for obtaining CST is using temperature-sensitive culture plates with poly (N-isopropylacrylamide) (PIPAAm). Smart interfaces have been developed to control the attachment of cells by manipulating the temperature. This method allows cells to be detached from the culture surface without the use of proteolytic enzymes, while maintaining the integrity of adhesion proteins such as E-cadherin and laminin and preserving the ECM components secreted by the cells [125]. Other excellent example are the temperature-responsive polymer brush coatings (TRPBCs). There is a commercially available product—Nunc™ Multidishes with UpCell™ Surface—that allows the adhesion of the cells of the culture solution at high temperatures and the detachment of the entire cell sheet at a lower temperature [134].

## 6. Functionalization Strategies to Promote Bioactivity in Polymeric Supports

Most polymers are not very reactive, and surface modifications that generate different chemical groups of interest are needed, or bioactive compounds (proteins, peptides, growth factors, hormones, enzymes, or other regulators of cell behavior) are added. The methodology used can be physical adsorption, chemical immobilization, or encapsulation. The surface properties of an implantable device are of critical importance, since the first contact with the organism is mediated by the cell–biomaterial interface. For this reason, many research groups are studying the optimal chemical and physical configurations of the surfaces of new biomaterials to promote cell interaction and to produce efficient tissue engineering constructs. Instructional biointerfaces could determine the type of cell to attach to the polymer and direct their behavior through the scaffold. The common purpose of surface treatment is to modify the outermost layer of a polymer to improve properties such as its wettability, sealability, adhesion to other materials, and interaction with a biological environment while keeping the mechanical properties of the polymer intact. When a scaffold is modified, the surface, chemical, and topological interactions that usually occur drive subsequent cellular and tissue events such as cell adhesion, cell orientation, cell motility, surface antigen display, cytoskeleton condensation, tyrosine kinase activation, cell modulation, and modification of intracellular signaling pathways that regulate transcriptional activity, gene expression, and inflammatory responses, etc. [34,40,135,136,137].

On the other hand, scaffolds can encapsulate and act as reservoirs for soluble bioactive molecules such as cytokines, growth factors, chemokines, or hormones that are subsequently released and act in a paracrine manner. In addition, the development of intelligent biomaterials that are capable of responding to specific stimuli such as pH, electrical signals, light, temperature, and metabolites e.g., adenosine triphosphate (ATP), and glucose is being explored. Such properties can be used to control cell adhesion, drug release, phase behavior, and mechanical parameters such as electrical conductivity, permeability, and tissue volume [138].

In conclusion, determining the design characteristics of a scaffold poses a great challenge, and it must be considered at the chemical-molecular level, taking into account the planned specific replacement therapy. Currently, the design of biomaterials can also be achieved by combining computer-aided design and nano-fabrication techniques for extreme precision design [8].

In Table 2, we tabulated examples of the polymers that have been subjected to both physical and chemical surface treatments and those in which biomolecules have been encapsulated for controlled release and that have thus achieved a more efficient scaffold.

## 7. Cells for Tissue Engineering: Stem Cells

Stem cells have the ability to self-renew and resume their undifferentiated state and in turn produce specialized progeny under the appropriate physiological conditions. These properties of stem cells present them as a crucial strategy for tissue regeneration [154,155]. Self-renewal is a characteristic fundamental to the survival of stem cells. Stem cells can vary their differentiation potential depending on the organ where they are found. In any case, there must be a regulated balance between cell self-renewal and differentiation [156]. An important concept to keep in mind is the cell niche. The cellular niche is made up of signaling molecules and other cells that perform nutritional functions that are responsible for maintaining tissue homeostasis [157]. Stem cells are extremely sensitive to the physical nature of their environments, and the cell niche is extremely important not only in directing the development and behavior of stem cells, but also in maintaining their state of differentiation [158]. Knowledge and understanding of the environment of stem cells is essential if one wants to recreate biomimetic structures seeded with stem cells in TE.

## 8. Stem Cell Classification

Table 3 shows the different types of stem cells classified according to their differentiation potential.

## 9. Cell Replacement Therapies Combined with Tissue Engineering Strategies: More Complex Than Is Believed

Cell therapy encompasses a broad clinical spectrum in which transplanted cells are applied in disease processes, such as cardiovascular disease, neurodegenerative disease, diabetes, autoimmune disorders, and cancer. In the last decade, there has been an exponential advance in the treatment of hematological malignancies, with promising preclinical results that produce a substantial clinical benefit. Despite this, other cell therapy approaches have not been successful, and this can be attributed to poor cell adhesion and survival at the site of injury [167]. It is here that TE plays a fundamental role, being able to promote cell regeneration using materials that serve as templates for cell and tissue growth and other approaches that rely on exogenous cells that have been implanted to become part of a pre-designed device [168].

As seen above, the mechanical properties of a scaffold can exert a significant influence on seeded stem cell differentiation, and also on the behavior of many mature cell types such as epithelial cells, fibroblasts, muscle, and neuronal cells. These are capable of preserving the rigidity of the substrate and displaying different morphological and adhesive characteristics against it [138]. The cells are capable of invading the matrix and organizing themselves in the form of function tissue by proliferating and migrating through the matrix architecture. This reorganization is followed by ECM synthesis to regenerate damaged tissue [168].

In the last decade stem, a cell science revolution emerged with the advent of human pluripotent stem cell (hiPSC) technologies and, more recently, the CRISPR/Cas9 genetic engineering technique and organoid technologies. The scope of this field is intended to encompass biomimetic engineering, to provide and design cells, tissues, and organs for the precise study of numerous diseases in the near future [169]. Recently, genetically modified stem and vascular cells were generated by improving their efficacy and safety from editing longevity genes and tumor suppressors. This same approach could be used to activate tissue regeneration by optimizing the study of cellular senescence and regeneration pathways [170]. An example of this is the use TE vascular grafts as an attractive alternative for creating blood vessels in vitro that maintain their function after the implantation. Generally, grafts have been manufactured using vascular smooth muscle cells (VSMC), but these cells are limited in quantity and proliferation potential. In recent years, numerous researchers have optimized the method of obtaining VSMC from the genetic engineering of IPSc [171].

In cell therapies, cells are isolated from a donor and transplanted into a recipient (Figure 15). When the donor and the recipient are the same, the procedure is referred to as an autologous transplant, and, when they are different individuals, the transplant is allogeneic. However, in vitro findings have not always been replicated in humans. Several reports note that many therapies appeal to potentially vulnerable individuals, raising ethical and safety concerns about using cells in an unregulated manner [172]. In the case of iPSCs generated from human cells, there are major hurdles preventing the use of these cells in the development of safe and efficient therapies, since there is a risk of tumor formation [161]. Therefore, a critical point is to guarantee a complete and irreversible differentiation of the stem cells to the desired progenitor cells or terminal target cell type. This could be possible if the appropriate trophic factors are added to the culture medium or incorporated as reservoirs in the scaffolds so that they are released in a controlled manner [18].

Next-generation stem cell therapies also face manufacturing and regulatory hurdles, as the cells are exposed to complex and sophisticated genetic engineering modifications, which are costly and require skilled labor. This is because the product must be customized and designed for each individual patient in appropriate facilities to allow viral transduction or gene editing. For large-scale commercial therapies, manufacturing systems must be optimized to generate commercial-scale quantities of cells without altering their therapeutic properties, or to generate engineering components for product(s) that are stable over time [164].

Cellular behaviors are so complex that it is necessary to have a thorough understanding of the mechanisms of morphogenesis and normal wound healing before designing a scaffold that will be combined with cell therapy, since cells use preprogrammed information and signaling to create or recreate functional structures [8]. For this, a critical point is to know what characteristics of biomaterials can affect the components of the cellular microenvironment to allow the design of scaffolds that act as extrinsic regulators of stem cells’ fates and regulate their differentiation towards the desired tissue [154].

Overall, it can be seen that there are still many limitations when choosing a suitable cell therapy for tissue regeneration. On the other hand, it is necessary to combine these therapies with TE strategies, where the immobilization of endogenous cells from a patient must be carried out in porous matrices after accounting for exhaustive selection criteria, and, finally, the device must be directed to where it can exert a beneficial effect for a given pathology [173].

## 10. Applications of Polymers for Tissue Engineering in Experimental Models

As described above, three basic elements are required in TE: bioactive scaffolds, cells with a compromised differentiation potential, and adequate biochemical signals that promote cell differentiation and angiogenesis. Furthermore, for the successful design of the scaffold, different techniques can be evaluated for use in its manufacturing. Next, reviews of several interesting strategies in eight main TE application areas, epithelial, bone, uterine, vascular, nerve, cartilaginous, cardiac, and urinary tissue, will be made with the aim of learning more about current approaches in TE.

### 10.1. Epithelial Tissue Engineering

The skin is the largest and fastest growing organ in the body. It plays a vital role in protecting the body against external damages from common skin injuries such as abrasions, burns, or lacerations. Traumatic wounds such as third degree burns and defects in the dermis layer are more difficult to treat due to limited self-healing capacity. Therefore, bioengineered human skin donors or constructs are required to support the regeneration process. However, skin regeneration remains a major challenge due to the scarcity of donor skin and poor long-term viability [174]. For skin regeneration, TE has used techniques ranging from simple autologous epidermal sheets to more complex double-layer skin substitutes such as grafts to achieve rapid wound closure and an aesthetic and functional scar. It must be porous to allow the diffusion of water, nutrients, and waste, and to promote cell migration, and have mechanical properties similar to those of the native tissue, with an adequate rate of degradation that matches the rate of regeneration. Finally, the scaffold must prevent bacterial infection. To lead to the regeneration of the dermis and epidermis, polymers with biomimetic properties are required, containing chemical signals, soluble growth factors, and physical and mechanical properties similar to the ECM of the native tissue [175].

Bayat et al. manufactured chitosan nanofibers containing bromelain at 2 and 4% *w*/*v* by the electrospinning method. Bromelain, a mixture of proteolytic enzymes present in all the tissues of the pineapple (*Ananas comosus*), has application in the treatment of wounds and inflammations, and it is used as a debriding agent in burns. In this study, the physicochemical properties of the nanofibers (viscosity, electrical conductivity, tension, swelling test), the drug release profile, enzymatic activity, cytotoxicity, and the potential of the synthesized fibers were evaluated. According to the results obtained, the chitosan-bromelain (Ch-Br) nanofibers at 2% *w*/*v* presented better physicochemical properties, a more accelerated release profile, and lower cytotoxicity than the 4% *w*/*v* counterpart, for which they were selected for in vivo studies. In conclusion, it was observed that Ch-Br nanofibers accelerate the recovery process in burn injuries in animal models (Figure 16). Therefore, this system has the potential to be proposed as a therapy to heal burns in humans [176].

### 10.2. Bone Tissue Engineering

Bone is a complex and dynamically active tissue in the body. It has the natural ability to regenerate when minor damage or injury occurs. When fractures and injuries do occur, long periods of time are usually required for recovery. Traumatic bone defects are considered a significant socioeconomic burden, primarily because these severe bone injuries lack the ability to fully reconstitute collapsed bone. For example, in the area of periodontics and maxillofacial surgery, there is a growing need to regenerate alveolar bone that has been lost due to chronic diseases, malignant neoplasms, or trauma. The demand occurs from patients seeking to prolong the longevity of their natural teeth or implant assisted prosthetics. Importantly, the regenerative capacity of injured bone is diminished due to the absence of a template for the regeneration of new tissue. To overcome this problem, the field of bone TE has advanced in the design of efficient therapeutic mechanisms that include the incorporation of cells, biomaterials, and the corresponding growth factors to promote osteogenesis, osteoinduction, and osteoconduction, as well as bone mineralization [177,178].

H. Samadian et al. manufactured a 3D scaffold based on the following components: poly (l-lactic acid) (PLLA)/Polycaprolactone (PCL) that act as a polymeric matrix, gelatin nanofibers (GNFs) that mimic the collagen fibers of the bone ECM, and gold nanoparticles (AuNPs) to mimic hydroxyapatite crystals and act as a healing agent for application in bone TE. To carry out cytotoxicity studies, MG-63 cells were cultured on the scaffolds, and it was observed that the greatest proliferation occurred in those membranes that contained PCL/PLLA/GNF/AuNPs (80 ppm) at 72 h. AuNPs are considered non-toxic and biocompatible structures at optimal concentrations. Regarding the in vivo tests, the samples with PLLA/PCL/GNF/AuNPs (80 ppm) induced the highest bone regeneration (Figure 17). Therefore, it was concluded that the combination of 3D scaffolds containing GNFs and AuNPs could mimic the native structure of the tissue and promote the healing process of the bone [179].

### 10.3. Urinary Tissue Engineering

Various reconstructive approaches have been used to treat and restore function in congenital and acquired pathologies that can affect the human urinary tract, such as hypospadias, strictures, fistulas, trauma, and cancer. However, there are certain factors that complicate the repair. Due to the poor quality of local tissues, additional tissues must be used for replacement, including genital skin and buccal and lingual mucosa. As these are available in limited quantities, their use is restricted and alternative approaches need to be sought [180]. Furthermore, mimicking urinary tissue is not easy due to the complexities of the native system. The field of urology has recently turned to TE and regenerative technology, and, although it is still in its infancy in clinical applications, a great deal of research is under way in search of solutions for urinary reconstruction [181].

Adamowicz et al. proposed a new composite biomaterial derived from the amniotic membrane (Am) covered with a layer of graphene. Graphene creates an interface between cells and external stimuli that replaces the neural network in the bladder. The main objective is to obtain a biomaterial with response to electrical stimuli for the application of urological TE in neurogenic bladder. In this case, porcine-derived smooth muscle cells (SMC) and urothelial cells (UC) were used to evaluate the properties of the developed biomaterial (Figure 18). The presence of the graphene layer significantly increased the electrical conductivity of the biocomposite. UC and SMC showed an organized growth pattern on graphene-coated surfaces and contractile response of SMC was observed when electrically stimulated [182].

### 10.4. Uterus Tissue Engineering

In mammalian reproduction, the uterus fulfills essential complex biological functions. These functions can be affected by congenital anomalies or acquired diseases that affect the integrity of the uterus. This can result in the woman’s inability to conceive or carry a viable fetus to term. Allogeneic uterine transplantation is a possible treatment strategy, but it requires donors and the use of anti-rejection therapies. Regenerative medicine and TE technologies become attractive options to overcome these limitations [183].

Magalhaes and collaborators described the design and fabrication of uterine tissue in vitro and its subsequent application in a rabbit model, performing a subtotal uterine reconstruction. They fabricated PGA/PLGA scaffolds in semicircular shapes and successfully seeded autologous myometrial-derived cells on the outer layer of the scaffold, and endometrial-derived cells on the interior of the scaffold using a sequential seeding method. To carry out the implantation, three experimental groups were taken into account: 1—rabbits with a subtotal semicircular excision in their remaining uterine horn that received the scaffold with cells, 2—those that received an acellular scaffold, and 3—those in which they were sutured the remaining uterine margins.

To investigate the in vivo functionality of the engineered uterine tissue, reproductive studies were carried out using fertile males for natural mating six months after graft implantation. Only the group that received the scaffold with cells, developed structures similar to native tissues (organized luminal/glandular epithelium, stroma, vascularized mucosa, and two-layer myometrium). Furthermore, it was compatible with pregnancy and the fetus had a normal development within the length of the segment where the reconstruction was carried out, which resulted in well-formed offspring with body weights similar to normal rabbits (Figure 19) [183].

### 10.5. Vascular Tissue Engineering

The functions of the organs are maintained by the vascular systems, which are responsible for the exchange of substances (nutrients, oxygen, and transport of waste products). The construction of channels and capillaries similar to the real ones is a great challenge, since they must promote the exchange of substances in vitro and, at the same time, preserve their structure and function [184]. In addition, the cells of the blood vessels have a complex structure and are functionally dynamic, but with minimal regeneration potential. The various functions they perform are due to the presence of a complex extracellular matrix that varies in composition, thickness, and general architecture, and the function differs between arteries, capillaries, and veins. For all these reasons, it is ideal to find a consistent approach in choosing appropriate materials for tissue-engineered vascular grafts [153].

Yilin Wang et al. fabricated an asymmetric two-layer scaffold by co-electrospinning poly(e-caprolactone) and chitosan (PCL/CS) and poly(e-caprolactone) with carboxymethylchitosan (PCL/CMC). Chitosan and carboxymethylchitosan were selected as the antibacterial and antithrombotic components, respectively. The inner layer corresponds to PCL/CMC, while the outer layer to PCL/CS (Figure 20).

To assess blood compatibility, the hemolysis ratio was investigated using red blood cells on the PCL nanofibrous scaffolds. All hemolysis rates were less than 2%, indicating that the scaffold could be safe for clinical use. Furthermore, the nanofibers showed good compatibility with red blood cells, since the morphology of those cells does not change, apparently. The anticoagulant activity of the grafts was also evaluated as a function of coagulation times. The inner layer exhibited thrombosis inhibitory activity, because the structure of CMC is similar to that of heparin, and the presence of carboxyl groups seems to prolong coagulation times and could effectively prevent restenosis. In addition, the whole blood circulation was evaluated, and the results showed that the asymmetric nanofibrous scaffold has good blood compatibility and can inhibit thrombus formation. On the other hand, only the PCL/CS outer membrane exhibited good bactericidal properties against *S. aureus*, and *E. coli*, indicating that this vascular graft could reduce the risk of postoperative infection.

To assess cell viability and adhesion, they seeded HUVEC on the nanofibrous membranes, and the results showed that the cells have good affinity for the scaffold. Furthermore, PCL/CMC nanofibrous membranes could enhance rapid endothelialization after implantation.

In conclusion, the researchers designed and fabricated an asymmetric bilayer scaffold with biodegradable properties, excellent mechanical properties, and good blood compatibility. Therefore, these artificial grafts show promise as substitutes for native blood vessels and could reduce the rate of restenosis and the risk of postoperative infection [185].

### 10.6. Cardiac Tissue Engineering

Despite significant advances in medicine and technology, heart failure is still associated with a high mortality rate throughout the world. Therefore, it is essential to apply new concepts for the development of new treatments and therapeutic alternatives [186].

Myocardial infarction is a very common heart disease worldwide. For that reason, cardiac patches gained great interest as therapeutic candidates to stimulate myocardial regeneration after their implantation. Thus, Fiamingo et al. studied Chitosan-based hydrogels implanted onto the epicardial surfaces of the infarcted myocardium of rats. The result revealed that chitosan hydrogels prepared from 1.5% polymer solutions were perfectly incorporated onto the epicardial surface of the heart and presented partial degradation accompanied by mononuclear cell infiltration [187]. Later, in order to determine the impact of the acetylation degree of chitosan in the degradation and biological activity, acellular chitosan hydrogels with different degrees of acetylation were implanted in a murine model of both ischemic and non-ischemic cardiomyopathies. This study demonstrated the beneficial effects of chitosan hydrogels, particularly with polymers with an acetylation degree of 24%, on cardiac remodeling in two cardiomyopathy models [188].

On the other hand, Lee et al. designed and fabricated different components of the human heart with complex collagen scaffolds at various scales, from capillaries to the entire organ. To achieve this, they used 3D bioprinting using the “FRESH” platform (freeform reversible embedding of suspended hydrogels). They managed to control the degree of gelation by adjusting the pH to obtain microfilaments and a porous structure that favor microvascularization, cell infiltration, and mechanical resistance for the fabrication and perfusion of multi-scale vasculature. Thanks to microcomputed tomography, hearts can be bioprinted more precisely, resembling the specific anatomical structure of each patient. In this study, they succeeded in printing cardiac ventricles using human cardiomyocytes, and these showed directional action potential propagation, synchronized contractions, and wall thickening during systole. Although the human heart was used as a proof of concept in this work, the collagen FRESH v2.0 platform could have potential for scaffold fabrication and applications in other tissue and organ systems. This study shows an important advance in the manufacture of scaffolds with structural, mechanical, and biological properties similar to those of native tissues, but there are still many challenges to overcome in order to reach the stage of clinical application, since the 3D bioprinting of a fully functional organ has not yet been achieved [189] (Figure 21).

### 10.7. Cartilage Tissue Engineering

There is an increase in degenerative joint diseases (e.g., osteoarthritis) as the general population continues to age. Overweight and sports injuries may occur in young and healthy people in whom spontaneous cartilage repair is limited. This is why there has been an increase in the demand for regenerative engineering approaches to cartilage. One of the causes could be the lack of vascularization of the articular cartilage, which would not allow an inflammatory response after the injury and the resulting repair. Another reason would be the intrinsic inability to self-repair due to the low proliferative capacity of chondrocytes, resulting in scar tissue with inferior mechanical properties and durability [31].

Gul Kim et al. designed and fabricated an artificial trachea with mechanical properties similar to the native trachea. For this, a two-layer tubular scaffold was fabricated: the inner layer with electrospun PCL nanofibers and the outer layer of 3D-printed PCL microfibers. To enhance the regeneration of the tracheal mucosa and cartilage, they used human induced pluripotent stem cells (iPSCs), human bronchial epithelial cells (hBECs), iPSC-derived mesenchymal stem cells (iPSC-MSCs), and iPSC-derived chondrocytes (iPSC-Chds) (Figure 22). Using a bioreactor system, the artificial tracheae were transplanted into a rabbit with a tracheal defect (1.5 cm in length). Endoscopy did not reveal granulation growth in the tracheal lumen. Alcian blue staining clearly showed the formation of ciliated columnar epithelium in iPSC-MSC clusters. Furthermore, micro-CT analysis showed that the iPSC-Chd groups formed neocartilage at the defective sites. Therefore, this study describes a promising long-term approach in the functional reconstruction of a segmental tracheal defect [31].

### 10.8. Neural Tissue Engineering

The nervous system is the most complex in the human body. When damage occurs, the motor and sensory functions are affected. Injuries to the central nervous system (CNS), which consists of the brain and spinal cord, generally lead to permanent disability, as spontaneous tissue regeneration is limited, and this leads to considerable socioeconomic problems [190]. The CNS can be affected by certain disorders, such as neurodegenerative diseases and ischemic injuries. While there are numerous approaches and treatments, very few result in a complete recovery or resolution. It is important to reduce the pathogenesis of the ongoing disease and prevent further tissue damage and recurrence [191]. In this regard, TE scaffolds are an attractive approach to replace damaged neural tissue or promote the regeneration of existing tissue to regain the functionality of the injured nerves [192].

A promising approach is the use of neural stem cells (NSCs), as they can differentiate into both neurons and glial cells and secrete neurotrophic factors, and their use does not raise the safety concerns of pluripotent stem cells. The combination of NSCs with biomaterials can improve the availability of cells at the site of injury and can fill the volume of the lesion cavity. The cells are protected from immunological attack, and axonal elongation is favored by providing adhesion and regulation signals to reduce inflammation and scar formation. Kourgiantaki et al. evaluated grafts based on collagen-based porous scaffolds (PCS) similar to the biomaterials used in clinical regenerative medicine already approved by the FDA. For this, 3D cultures of neural stem cells (NSCs) were combined with cylindrical collagen structures, and the graft was implanted in an animal model after generating a crush injury in the spinal cord. The evidence suggests that this type of support protects and regulates the cellular activity of the NSCs at injury sites, since PCS grafts seeded with NSCs favor the modulation of key processes such as cell differentiation, axonal elongation at the site of lesion and across the lesion boundary, the migration of NSC to the surroundings, and the reduction of astrogliosis. In addition, the in vivo functional integration of the implant was observed by means of histological tests. On the other hand, the results show improvements in recovery of locomotion in mice after 12 weeks of injury, and their locomotion performance does not differ statistically from that of uninjured control animals. These findings demonstrate the potential of PCS combined with NSCs as future therapies to induce nerve tissue regeneration in spinal cord injuries (Figure 23) [193].

### 10.9. Adipose Tissue Engineering

Adipose tissue is one of the most abundant tissues in the human body, but its regeneration in vitro and in vivo still presents substantial challenges. Currently, autologous fat grafting is used, but necrosis-induced volume loss occurs in the long term. One strategy is the use of primary cells and cell lines seeded on synthetic and natural polymer spheroids or scaffolds for the regeneration of adipose tissue [194]. In addition, adipose tissue is an important source of mesenchymal stem cells, and this makes it an interesting alternative for use in tissue engineering and cell regeneration [195].

Rodriguez et al. evaluated two types of scaffolds with different microstructures, one made of fibrous structured atelocollagen (FSA) and the other made of honeycomb-structured atelocollagen (HCA). The aim of the study was to carry out in vivo adipogenesis induction by implanting the scaffolds in four-week-old male severe combined immunodeficiency (SCID) mice. To evaluate the results, histological and immunohistochemical tests were performed after one, two, and four weeks of implantation. After these procedures, it was verified that the FSA scaffold had a greater capacity to induce adipogenesis in vivo without the incorporation of preadipocytes or adipogenic factors, while the HCA scaffold induced a severe acute inflammatory response. Therefore, it appears that the fibril organization of the FSA scaffolds is a key factor in the development of the adipogenic response. In addition, it was possible to obtain primary adipose tissue in the FSA scaffolds, composed of a mixture of white and brown adipocytes at week 2, and only white adipocytes at week 4. After carrying out immunostaining with a positive Pax7 result at weeks 1 and 2, the authors suggested the existence of a common myogenic progenitor shared by the observed brown and white adipocytes. For all this, the FSA scaffolds are presented as a promising structure for the engineering of brown and white adipose tissue (Figure 24) [196].

## 11. Tissue Engineering Polymers in Clinical Applications

The objective of TE is to continue expanding as a scientific platform to advance in the regeneration and repair of different types of tissues. For certain products to reach the clinical phase, where applications and studies are carried out in human beings, they must go through a rigorous control process after successful results have been obtained in preclinical studies. Currently, modern medicine uses polymer-based implantable medical devices for diagnosis or regenerative therapies in various clinical applications for tissue repair, and biomaterials often cause an immediate response in the host after being implanted that includes tissue injury, inflammation, cell proliferation, and tissue remodeling. The host response affects the degradation process, which results in changes in the structure, morphology, and mechanical properties of the scaffold, and it is reflected in a decrease in yield. To develop functional and biodegradable devices for the replacement or repair of damaged tissues, a deep understanding of the degradation process of the material and the specific interaction between it and the host system is required [69].

There are important factors to take into account, such as differences in the age, nutritional status, physical activity, and disease status of patients, since, for a bioengineered tissue to be clinically relevant, it needs to integrate seamlessly into the body after implantation in vivo and play an active role in the development of tissue repair [28]. Thanks to specific biological and technological advances, it is possible to design and build tissues with a predictive performance based on previously identified patient needs. This will result in high quality and robust TE product under regulatory approvals. At the same time, they become attractive products for investors, since, by reducing the risks associated with product failure in the market stage, they become a new sector associated with medicine capable of revolutionizing the future of health care [197].

Advances in this area can be found at ClinicalTrials.gov, a database of publicly and privately funded clinical trials conducted around the world. Most of the records on ClinicalTrials.gov describe clinical trials in which human volunteers are assigned to interventions (for example, a medical product, behavior, or procedure) based on a protocol, and then their effects on the biomedical or health outcomes are evaluated. Table 4 briefly describes some approaches used in clinical trials using composite scaffolds as starting material.

## 12. Tissue Engineering Polymers in the Market

Costs related to organ replacement represent 8% of global health spending, and it is estimated that, by 2040, 25% of US GDP will be linked to health. The developments and progress in the design and engineering of biomaterials have made possible a new generation of attractive materials for regenerative medicine, but a question arises: which of these materials will successfully position itself in the market? That will depend on a combination of clinical results, performance, commercialization, and profitability. The materials used must contain complex information encoded in their physical and chemical structures, since one drawback is their ability to influence cell behavior. On the other hand, complexity must be minimized to obtain financial profitability. There should be several phases of product development, beginning with the identification of the simplest functional performance required to solve a defined clinical problem. The initial ambitious goals of rebuilding entire organs have largely given way to smaller, more achievable goals: for example, instead of trying to replace an entire heart, clinical advances in cardiac repair focus on manufacturing coronary arteries, valves, and myocardium [20].

Numerous companies have developed TE scaffolds of different materials and shapes that have reached the commercialization stage and can now be found as off-the-shelf products that have already undergone adequate testing and controls. The Nanotechnology Products Database (NPD) provides a trusted source of information on the nanotechnology products currently used in a wide range of industrial applications.

We can find products with nanofibers being used as artificial arteries, 3D biological inks, pre-loaded well scaffolds, or biocompatible polymers sheets as a replacement for traditional lab plasticware for tissue culture, custom 3D scaffolds for bioreactor scale up and regenerative medicine, or resorbable synthetic hydrogel composed of repetitive amino acid sequences of arginine-alanine-acid and aspartic-alanine prepared in aqueous solutions.

## 13. Conclusions

Herein, we provided a comprehensive overview of recent advances in the biological applications of biodegradable polymers in tissue engineering, ranging from basic scientific concepts to clinical application. Currently, different polymer-based medical devices have been approved for use in clinical trials, and a wide variety of polymeric biomaterials are available as commercial products.

At the same time, the choice of a good scaffold is critical, and the polymeric composites must exhibit properties such as the porosity, permeability, fibrousness, and mechanical stability necessary to successfully mimic extracellular matrices and provide the necessary structural support for the subsequent cell adhesion and tissue regeneration. Despite their many advantages, including biocompatibility, biodegradability, and others, these materials are rather inert and lack the specific functionalities that would endow them with additional biological and responsive properties. This is why numerous researchers are searching for new, smart functional materials that promote cell functions and tissue regeneration.

We are in an era of biological renaissance that poses multiple challenges to the scientific community in the areas of regenerative medicine and organ and tissue engineering. Despite great advances in science and technology, there is still a lot of research work ahead, and numerous efforts are currently underway to improve the mechanical properties of scaffolds with the aim of increasing the number of commercially and clinically successful trials. Knowledge and technology related to polymeric scaffolds are growing exponentially, and improvements in regenerative medicine will undoubtedly lead to safer and more integrated tissue-engineered systems to treat human disease. In the future, there could be “tissue and organ banks” that are available to any patient. To achieve this goal, regenerative medicine and tissue engineering technologies will require substantial and ongoing interdisciplinary research leading to successful results.

## Figures and Tables

**Figure 4 bioengineering-10-00218-f004:**
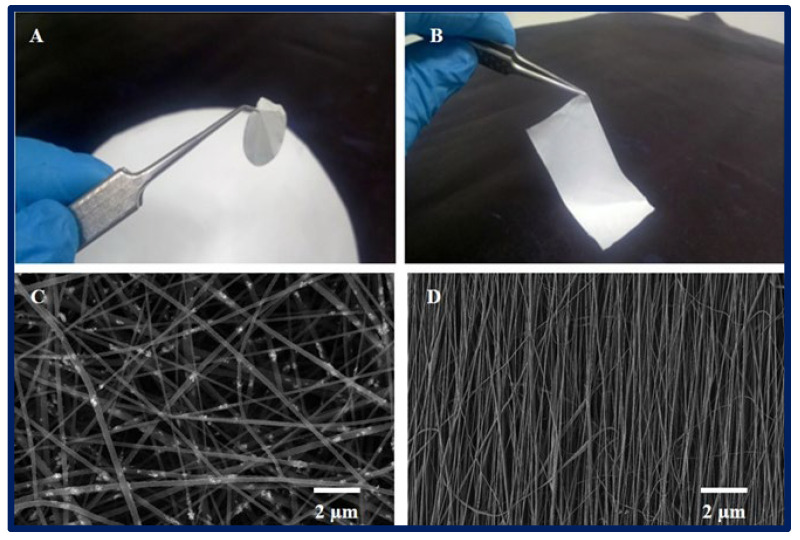
(**A**,**B**): macroscopic appearance of a 2D PCL scaffold fabricated by electrospinning technique. (**C**,**D**): microarchitecture of scaffolds observed under scanning electron microscopy, (**C**): randomly oriented nanofibers, and (**D**): nanofibers with parallel orientation.

**Figure 5 bioengineering-10-00218-f005:**
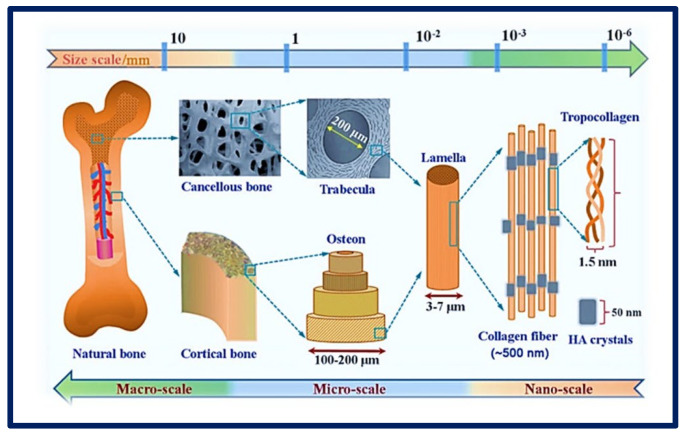
Complexity of an organ showing its macro, micro, and nanoarchitecture. Chemical composition and multi−scale structure of natural bone. Bibliography consulted [35] is licensed under CC BY 2.0. To view a copy of this license, visit http://creativecommons.org/licenses/by/4.0/ (accessed on 15 December 2022).

**Figure 6 bioengineering-10-00218-f006:**
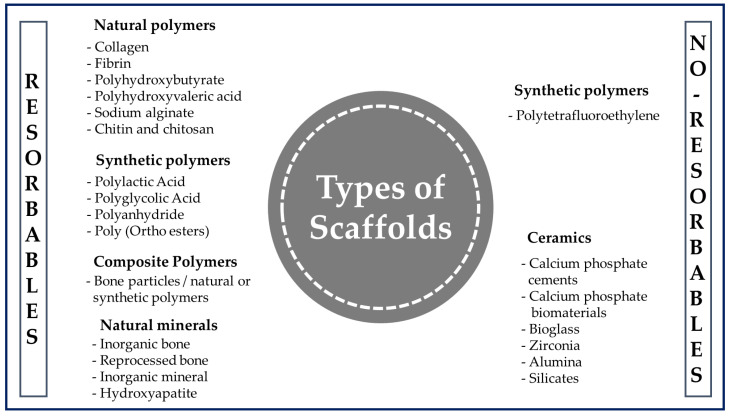
General classification of polymers for medical use.

**Figure 7 bioengineering-10-00218-f007:**
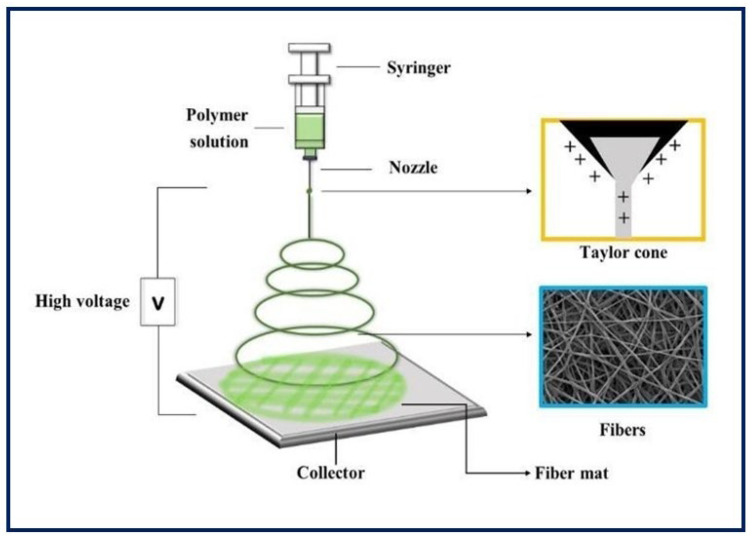
Basic electrospinning equipment.

**Figure 8 bioengineering-10-00218-f008:**
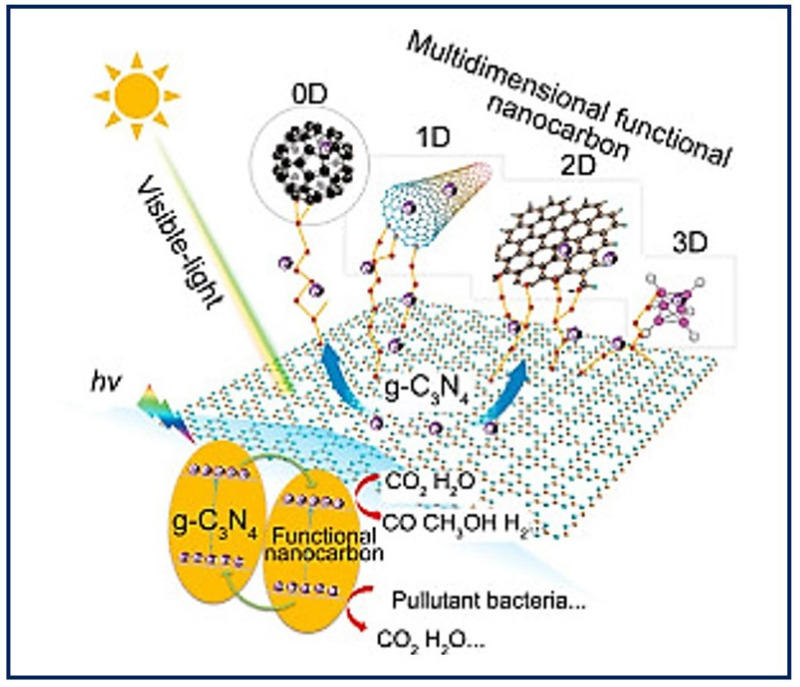
Utilization of multidimensional functional nanocarbon materials (0D–3D) as modifiers to enhance the visible light photocatalytic activity of g-C3N4. Bibliography consulted [82] is licensed under CC BY 2.0. To view a copy of this license, visit http://creativecommons.org/licenses/by/4.0/, (accessed on 15 December 2022).

**Figure 9 bioengineering-10-00218-f009:**
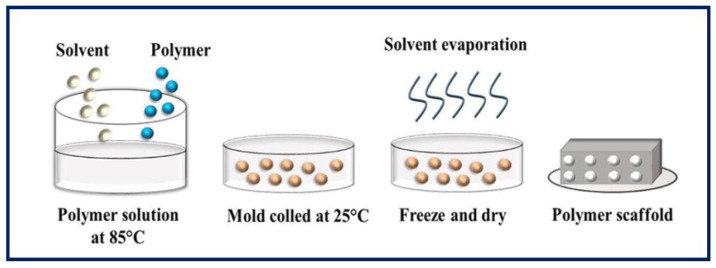
Schematic diagram of solid-liquid phase separation. Bibliography consulted [88].

**Figure 10 bioengineering-10-00218-f010:**
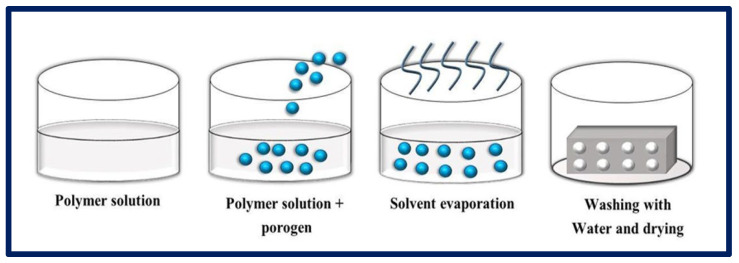
MM-PL process, where a polymer is mixed with a porogen to obtain porous scaffolds. Bibliography consulted [88].

**Figure 11 bioengineering-10-00218-f011:**
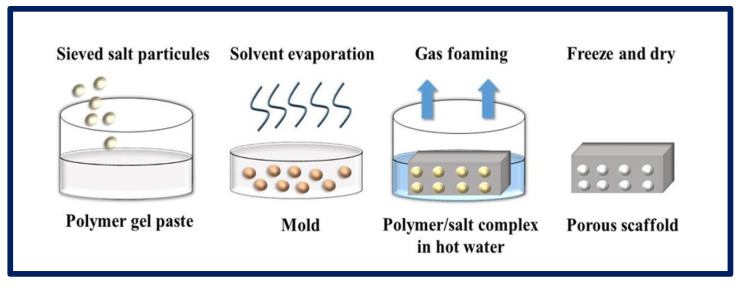
Gas formation process by chemical method for the production of porous scaffolds. Bibliography consulted [88].

**Figure 12 bioengineering-10-00218-f012:**
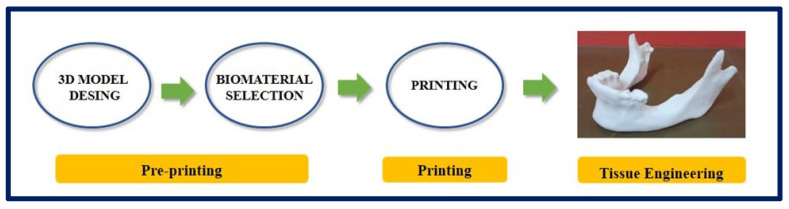
Process of design and manufacture of a jaw by means of 3D printing. Bibliography consulted [116].

**Figure 13 bioengineering-10-00218-f013:**
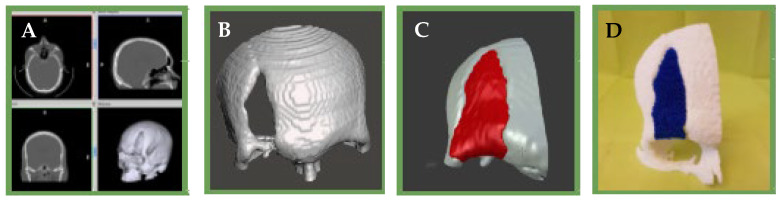
(**A**): Tomographic image of the patient’s skull (**B**): 3D reconstruction of the skull and the bone defect and surrounding tissue (**C**): 3D computational model of the bone defect (**D**): 3D impression of the scaffold and surrounding bone tissue.

**Figure 14 bioengineering-10-00218-f014:**
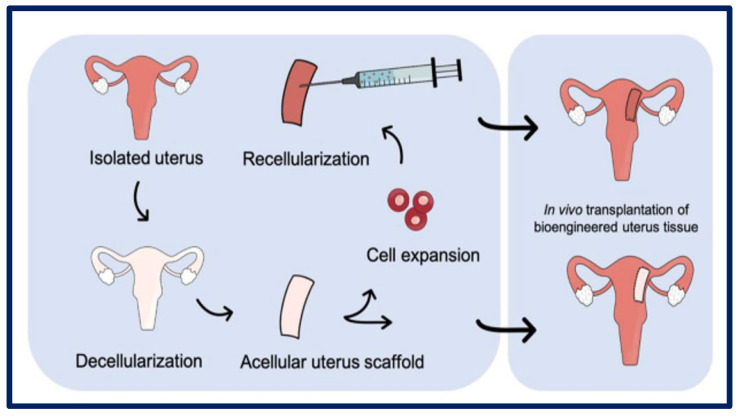
Uterus decellularization process and subsequent cellularization. Bibliography consulted [118] is licensed under CC BY 2.0. To view a copy of this license, visit http://creativecommons.org/licenses/by/4.0/ (accessed on 15 December 2022).

**Figure 15 bioengineering-10-00218-f015:**
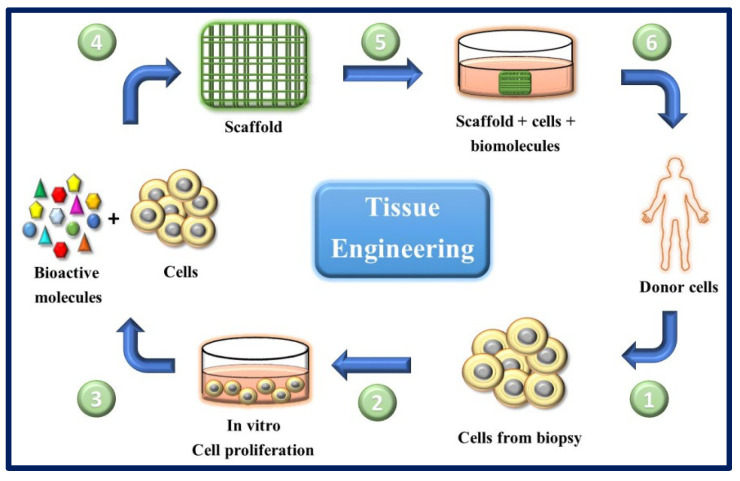
General scheme of the ideal circuit in regenerative therapies combining cells with appropriate molecules and biomaterials. (1) Obtaining cells from the patient (for example, from a skin biopsy). (2) In vitro culture and expansion of autologous cells.
(3) and (4) cells are seeded on scaffolds along with biomolecules of interest. (5) The scaffold-molecules-cells system is pre-incubated under optimal conditions. (6) The scaffold is implanted in the patient.

**Figure 16 bioengineering-10-00218-f016:**
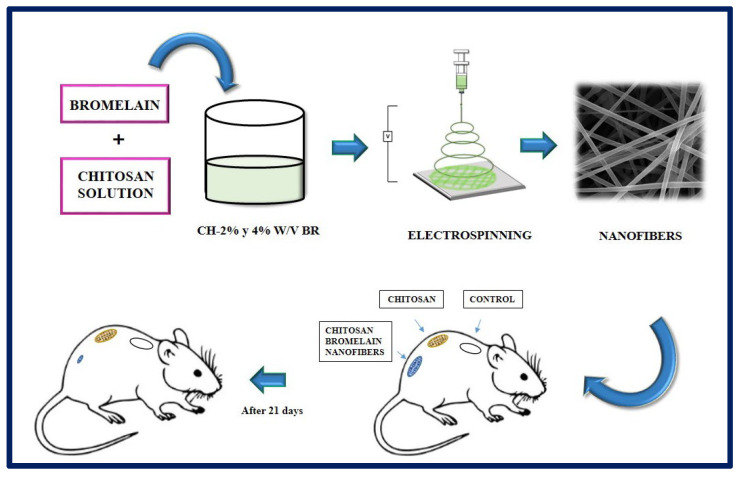
Chitosan nanofibers containing bromelain (2% and 4% *w*/*v*) were prepared by electrospinning method. The burn wounds of the rats were treated with chitosan nanofibers, and chitosan-Bromelalin nanofibers. The results indicated that chitosan-2% *w*/*v* bromelain nanofiber was more efficient to heal burn skin compared to chitosan nanofiber alone in the animal model tested. Bibliography consulted [176].

**Figure 17 bioengineering-10-00218-f017:**
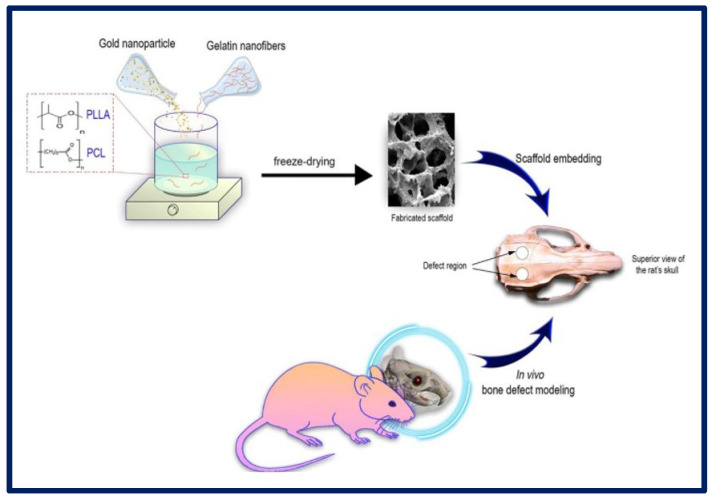
Schematic representation of the in vitro and in vivo experiments using PLLA-PCL-GELATIN scaffolds with gold nanoparticles for bone tissue regeneration. Bibliography consulted [179] is licensed under CC BY 2.0. To view a copy of this license, visit http://creativecommons.org/licenses/by/4.0/ (accessed on 15 December 2022).

**Figure 18 bioengineering-10-00218-f018:**
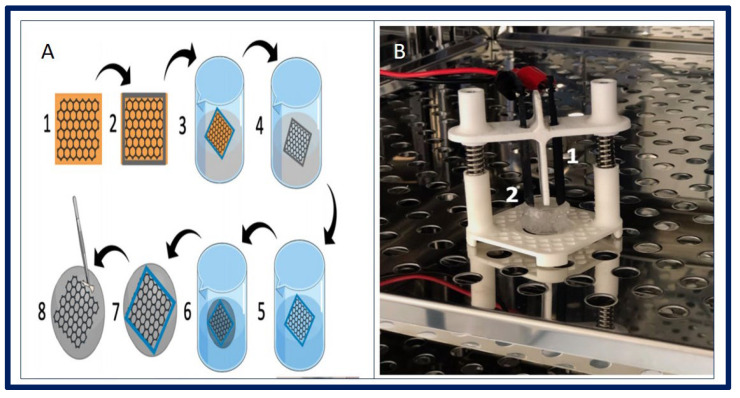
(**A**): fabrication of biocomposite. (1) Graphene layer on copper foil. (2) PDMS frame adjusted to desired biomaterial shape. (3) Etching Cu foil by ammonium persulfate solution. (4) Floating graphene layer in PDMS frame. (5) Washing out ammonium persulfate. (6) Fishing graphene layer posited in PDMS frame with Am. (7) Graphene placed on Am surface. (8) Carful mechanical removing of PDMS frame. (**B**): 3D-printed stimulator in CO2 incubator. (1) Graphene based electrodes (2) Biocomposite fixed in cell crown. Bibliography consulted [182] is licensed under CC BY 2.0. To view a copy of this license, visit http://creativecommons.org/licenses/by/4.0/ (accessed on 15 December 2022).

**Figure 19 bioengineering-10-00218-f019:**
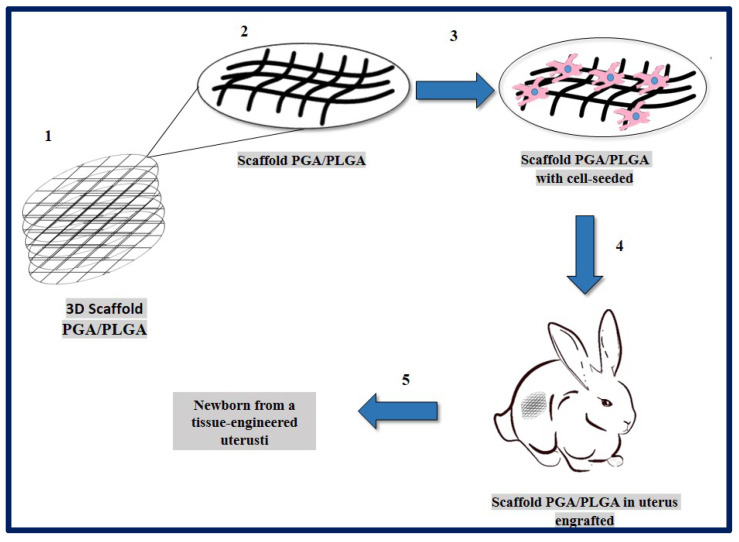
PGA/PLGA scaffold fabrication and in vivo implantation to tissue-engineered uterus. (1) PGA/PLGA scaffold macroscopic image. (2) PGA/PLGA scaffold details that shows its microstructure. (3) Scaffold SEM image with cell-seeded before implantation. (4) Scaffold engrafted to uterus tissue in a rabbit model. (5) Fetus was obtained then tissue-engineered uterus. Adapted from bibliography [183].

**Figure 20 bioengineering-10-00218-f020:**
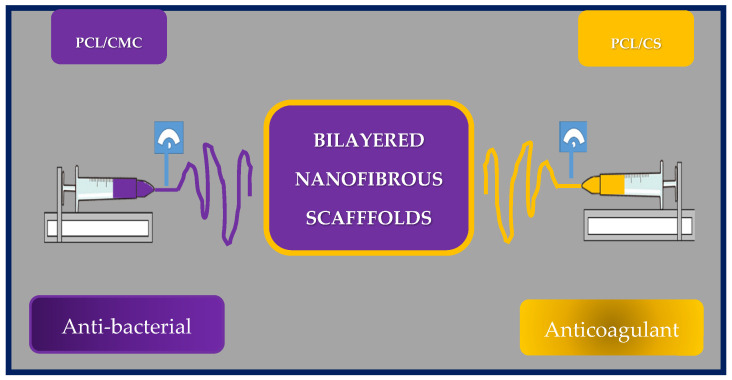
Schematic representation of the manufacture of asymmetric two-layer scaffold by co-electrospinning of poly(e-caprolactone) and chitosan (PCL/CS) and poly(e-caprolactone) with carboxymethylchitosan (PCL/CMC) scaffolds. Bibliography consulted [185].

**Figure 21 bioengineering-10-00218-f021:**
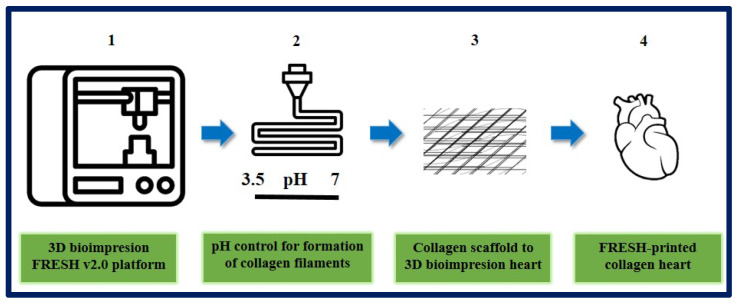
Organ-scale FRESH v2.0 3D bioprinting. (1) 3D bioprinting of collagen using FRESH v2.0 in high resolution. (2) pH control for rapid neutralization causing gelation to collagen filament formation process. (3) Single filaments of collagen obtained to this system were smooth with 20 to 200 nm diameter. (4) In the first instance, the ventricle was printed with a central section of cardiac cells, internal and external collagen shell, and a collagen-only section. Then Tri-leaflet heart valve 3D model at adult human scale was printed with the same system. Finally, a FRESH v2.0 collagen heart was printed at a neonatal scale. Bibliography consulted [189].

**Figure 22 bioengineering-10-00218-f022:**
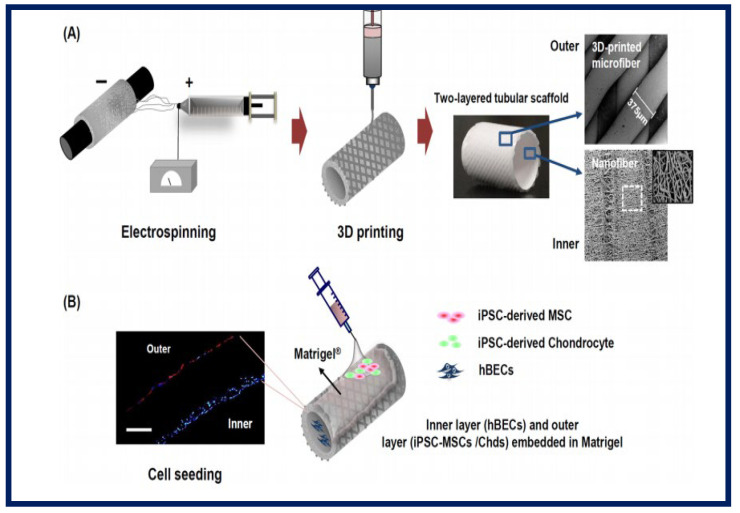
Illustrative schematic of the fabrication of three-dimensional tubular artificial tracheal scaffolds (**A**): The inner layer was fabricated from electrospun polycaprolactone (PCL) nanofibers and the outer layer from 3D-printed PCL strands. (**B**): Scaffolds were seeded with induced pluripotent stem cell-derived chondrocytes (iPSC-Chds in the outer layer and human bronchial epithelial cells (hBEC) in the inner layer of the scaffold. Identification was by PKH-26 staining (red color; iPSC-chds) and DAPI staining (blue color; hBEC). Bibliography consulted [31] is licensed under CC BY 2.0. To view a copy of this license, visit http://creativecommons.org/licenses/by/4.0/ (accessed on 15 December 2022).

**Figure 23 bioengineering-10-00218-f023:**
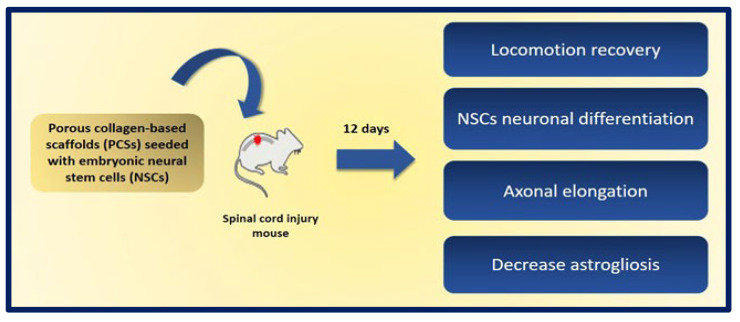
Graphic representation of the benefits obtained after 12 days of the implantation of porous collagen-based scaffolds (PCSs) seeded with NSCs in a murine model of spinal cord injury. Bibliography consulted [193].

**Figure 24 bioengineering-10-00218-f024:**
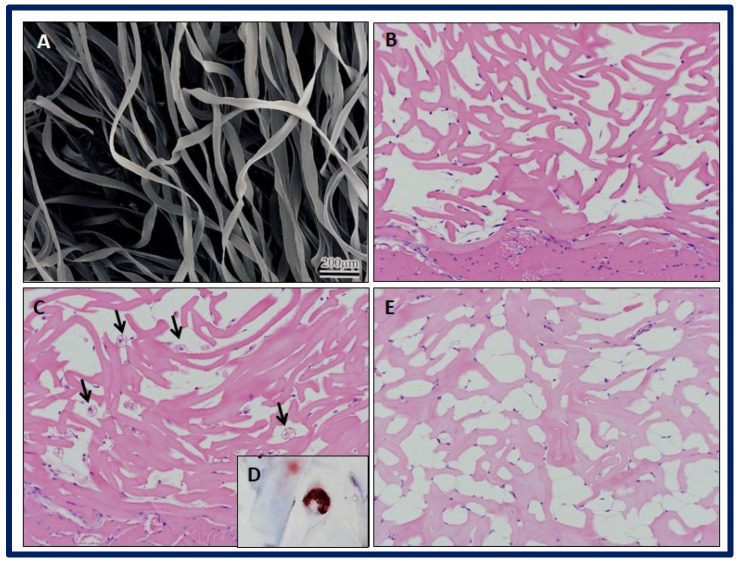
SEM micrograph showing the (**A**) FSA scaffolds. Histological examination of implant specimens of FSA scaffolds at different weeks (Hematoxylin and Eosin staining). (**B**) Week 1: Note the presence of stained spindle cells attached to scaffold fibers. (**C**) Week 2: Black narrows indicate brown-like adipocytes within the scaffold, with multiple cytoplasmic lipid droplets. Note the presence of some small spherical cells close to scaffold fibers. No inflammatory reaction is observed. (**D**) Week 2: Multiple cytoplasmic lipid droplets within the cells are stained with Oil Red O staining. (**E**) Week 4: Multiple white-like adipocytes within scaffold bulk are observed, with large cytoplasmic lipid droplets. Bibliography consulted [196].

**Table 2 bioengineering-10-00218-t002:** Different strategies for modification of scaffolds to promote their bioactivity.

Polymer	Molecule/Cell	Application	Technique	Ref.
PLGA scaffold	CellROX_ Green Reagent, and pHrodoTM Red AM (fluorophores)	Cell behavior study	Encapsulation	[139]
Hyaluronic acid-based hydrogel	Anti-Nogo receptor (anti-NgR) RNA aptamer	Spinal cord injury	Encapsulation	[140]
Chitosan oligosaccharide/ heparin nanoparticles	Stromal cell-derived factor 1 (SDF-1) and Bone morphogenetic protein 2 (BMP-2)	Bone tissue engineering	Encapsulation	[141]
Poly (ε-caprolactone) (PCL)/nano-hydroxyapatite	Mesenchymal stem cell-encapsulated in HPCH (hydroxypropyl chitin hydrogel)	Bone regeneration	Encapsulation	[142]
Ethylcellulose and polylactic-co-glycolic acid (PLGA)	Hemoglobin (Hb) and bovine serum albumin (BSA)	Encapsulation efficiency of proteins	Encapsulation	[143]
Hyaluronic acid (HA)/poly-L-lysine (PLL) layer-by-layer (LbL) self-assembly coating on β-TCP (β-tricalcium phosphate)	Small extracellular vesicles	Bone regeneration	Surface modification	[144]
Chromatin (DNA and histone) supramolecular fibers as scaffold	Murine brain-derived neural stem cells (NSCs)	Neural regenerative medicine	Encapsulation	[145]
3D-printed (3DP) calcium phosphate (CaP) scaffolds	Polydopamine with Cissus Quadrangularis extract	Treatment bone defects	Surface modification	[146]
Porous composite scaffold of chitosan, chondroitinsulfate, gelatin	Nano-bioactive glass (nBG) (60% SiO2, 36% CaO, and 4% P2O5)	Bone tissue regeneration	Encapsulation	[147]
Polyimide	Aminopropylmethacrylamide, Reactive succinimidyl ester, and Methacrylamide modified gelatin	Ocular diseases: age-related macular degeneration (AMD)	Surface modification	[148]
Silk fibroin microparticles	N-(2-aminoethyl)-4-(4-(hydroxymethyl)-2-methoxy-5 nitrosophenoxy) butanamide (NB)	Cartilage regeneration	Surface modification	[149]
Methacrylated-Hialuronic Acid	Basic Fibroblastic Growth Factor (bFGF)	Skin wound healing	Surface modification	[150]
Poly(glycolic acid)–poly(ethylene glycol)–poly(glycolic acid)-di(but-2-yne-1,4-dithiol) (PdBT)	Hydrophilic bone morphogenetic protein mimetic (BMPm) peptide Hydrophobic N-cadherin (NC) peptide Cartilage-derived glycosaminoglycan macromolecule, chondroitin sulfate (CS)	Mesenchymal stem cell (MSC) encapsulation	Surface modification	[151]
Porous 3D PLGA scaffold	Smoothened agonist sterosome	Bone regeneration	Surface modification	[152]
Poliuretano	REDV peptide	Improve hemocompatibility by promoting EC attachment, proliferation, and growth	Via an active p -nitrophenyloxycarbonyl group	[153]

**Table 3 bioengineering-10-00218-t003:** Classification of stem cells according to their plasticity.

Potentiality	Cell Type	Source	Features and Mature Cell Lineage
Totipotent stem cells	Embryonic stem cell (ESc)	Zygote [159]	A single cell capable of dividing and forming several differentiated cells. Cells including extraembryonic tissues [159].
Pluripotent stem cells	Embryonic stem cell (ESc)	Isolated from the inner cell mass of the blastocyst [160].	They give rise to any cell type of the three germ layers. They have the ability to grow indefinitely while maintaining pluripotency [161]. Both types of cells can give rise to teratomas [162].
	Induced pluripotent stem cells (iPSc)	Obtained by genetic modification of somatic cells such as fibroblasts, to which specific transcription factors were introduced to induce pluripotency [163].	
Multipotent stem cells	Adult stem cells	They can be isolated from various tissues, including bone marrow, adipose tissue, umbilical cord, and dental pulp, among others (e.g.,: mesenchymal cells, hematopoietic cells) [164,165].	They can give rise to a large number of cell lineages [164]. MSCs have immunomodulatory, anti-inflammatory, angiogenic, antiapoptotic, and trophic properties [165].
Oligopotent stem cells	Adult stem cells	They can be isolated from the blood as myeloid and lymphoid cells.	They can give rise to a limited number of cell types. Lymphoid stem cells can only differentiate into basophils, neutrophils, eosinophils, monocytes, and thrombocytes [159].
Unipotent stem cells	Adult stem cells	Epidermal, satellite (SC)	They can give rise to a single cell type. For example, SC are involved in skeletal muscle regeneration and are normally inactive until a stimulus or damage occurs and are activated to trigger the formation of new muscle fibers [166].

Refs. [159,160,161,162,163,164,165,166] Bibliography consulted.

**Table 4 bioengineering-10-00218-t004:** Different polymer-based medical devices approved to use in clinical trials.

Device	Description	Application	Identifier	Ref.
3D-Printed Scaffold	Polycaprolactone Tricalcium Phosphate (PCL-TCP) is a bioactive, biocompatible, and bioabsorbable non-toxic polymer compound.	Ridge preservation after tooth extraction.	NCT03735199	[198]
Neuro-spinal Scaffold	Poly (lactic-co-glycolic acid)-b-poly (L-lysine) scaffold	Thoracic AIS A traumatic spinal cord injury at neurological level of injury of T2-T12.	NCT02138110	[199,200]
Absorb GT1 BVS	Absorb GT1 Bioresorbable Vascular Scaffold (BVS)	Ischemic heart disease, angina pectoris, coronary artery disease, coronary artery occlusion, myocardial ischemia	NCT03409731	[201]
Bio ACL	Collagen based-membrane derived from amniotic tissue	anterior cruciate ligament rupture	NCT03294759	[202,203]
BMAC Nerve Allograft	Decellularized processed peripheral nerve allograft, with autologous bone marrow aspirate concentrate.	Peripheral nerve injury, upper limb	NCT03964129	[204]
Bioactive glass scaffold	Bioactive glass scaffold with multi-scale porosity prepared using the sol-gel technique.	Bone loss, vertical alveolar bone loss, horizontal alveolar bone loss	NCT01878084	[205]
ABSORB scaffold	Everolimus-eluting bioresorbable vascular scaffold	Cardiac allograft vasculopathy	NCT02377648	[206,207]
SERI^®^ Surgical Scaffold	Bioresorbable scaffold derived from silk, developed to provide support and repair of soft tissues	Breast reconstruction	NCT01256502	[208]
Chitosan scaffold	Bilaminar chitosan scaffold	Sellar floor repair in endoscopic endonasal transsphenoidal surgery	NCT03280849	[209]
Nanofiber scaffold	Rotium nanofiber is an FDA approved scaffold.	Rotator cuff tears	NCT04325789	[210]
Bioresorbable vascular scaffold	The bioresorbable vascular scaffold (BVS) has been approved and is used in daily clinical practice.	Coronary thrombosis, tomography, optical coherence drug-eluting stents	NCT03180931	[211,212]
Firesorb	Sirolimus Target Eluting Bioresorbable Vascular Scaffold	Coronary artery disease	NCT02890160	[213]

## Data Availability

Not applicable.

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
