# Peer review of "Polymeric Materials, Advances and Applications in Tissue Engineering: A Review"

_bioengineering, 2023, doi:10.3390/bioengineering10020218_

Round 1

Reviewer 1 Report

In this article " Polymeric Materials, Advances and Applications in Tissue Engineering: a review ", The author mentioned about the various kinds of polymeric materials and their applications. After reviewing this article, it can be considered for publication in this journal. However, there are some issues that have to be fixed before publication;

The abstract should be revised to shows the work more appropriately to increase the interest of readers.

Please revised introduction with proper consequences with some new references (Polymers 13 (1), 135, Polymers 14 (4), 845)

Some errors regarding the sub/super script, spacing and typo need to consider throughout the manuscript.

Please specify the novelty, significance, technical merit of the study in a better way.

The figure should be revised to high quality as some are in blur format.( Fig 1, 6)

Most of the figures are taken from previously published articles, please make sure about the permission, if needed.

English is very poorly presented throughout the manuscript. as well as the presentation is also very poor, should be uniform.

Results and discussions should be clear compared and a critical analysis of its technical feasibility and applicability for this material should be given.

Make sure that the format of references are uniform.

For Table 5, please make sure for the permission aspect to mention the details of the products related to any particular brand.

The conclusion must be concise.

Reviewer 2 Report

I cannot recommend this review to accept for publication since information about a lot of the modern approaches including smart biomaterials for tissue engineering was omitted in this study. 

As was described in https://doi.org/10.3390/polym14194245 "In the past, the first materials capable of interactions with bacterial and eukaryotic cells, tissues and proteins were intuitively chosen by scientists without ability to impact these objects in the controlled manner. The second generation of the materials for biomedical applications was essentially improved in comparison to the first one; the surfaces of these materials were often modified by substances that had no toxic effect on the objects studied. Very rarely have these materials had a controlled impact on the biological systems, which was realized mainly by tuning the chemical nature of the materials. At the present time, the new type of materials for biomedical applications with active remote impact on the biological object is developing and in some cases is included in medical protocols." Unfortunately, the authors concentrated attention only on the second generation of materials for tissue engineering. 

What about studies describing tissue-engineered cell sheets?

Fig. 1 should be corrected taking into account the smart biomaterials for tissue engineering. 

Please cite relevant papers to improve the quality of the review:

https://doi.org/10.3390/polym14194245

10.1056/NEJMoa040455

https://doi.org/10.1021/la036139f

https://doi.org/10.1007/978-1-0716-2261-2_15

https://doi.org/10.1016/j.colsurfb.2014.03.049

Reviewer 3 Report

 The article by Socci et al. provides an extensive review on the state of the art of polymeric materials as scaffolds for tissue engineering applications. The authors describe the main types of biomaterials, the principles of their fabrication, their pre-clinical and clinical applications, a s well as example of commercial products.

The topic of porous scaffolds is quite busy in the literature, even though most of the papers are focused on a specific type of scaffold and a comprehensive review is missing in the most recent years. Indeed, I appreciated how the authors systematically addressed all the relevant topics, with a general part first and additional sections dedicated to specific fields, such as skin tissue engineering, bone TE etc. However, I believe that the interest of the manuscript would benefit from some interventions before being accepted - see the specific remarks below.

In addition, even though I am not mother tongue, the manuscript has several typos and flaws in the use of the English language and definitely needs editing and English revision – I just cited a few examples below. Such an editing will also contribute to diminish the significant differences in the style and quality of the langue, currently present in the various sections, likely written by the different co-authors.

MAJOR REMARKS

1) Figures: several figures can be implemented, since in the current version they are not particularly illustrative.

Fig 1: I believe that the “early years”, although poignant, can be escaped as non-scientific and I would rather do the effort to double check for the first example of transplantation – it seems to me that skin graft are the very first to be attempted, even before of Tagliacozzi. The references to art can be preciously put in the text.

Fig 2: The figure is too simple to be actually useful. More importantly, the main message being on “porous” scaffolds, it does not vehiculate the concept of porous scaffold and its advantages – and the cells are indeed drawn on the surface of the construct, which is misleading.

Fig 3: Some of the keywords shown in the circles are not logically  at the same level: for instance, growth factors should be cited along with cytokines etc. rather than proteins and peptides, that are lore general categories; similarly for stem cells vs mesenchymal cells: I would either oppose embryonic stem cells to adult stem cells or else, but not all-inclusive SC with an embryonic tissue.

2) Line 141, “the scaffold needs to be vascularized and integrated”. This point is too critical to be left without at least one or two references to highlight the state of art: for instance, Joanne et al. 2016 in Biomaterials (PMID: 26708641)

3) Lines 148-158. The content and style of this paragraph (“ in this review we will put emphasis on…”) is out of scope in this section of the text. It is very preliminary and, if used, shall go in the Introduction section.

4) Lines 402 etc. , section: “4.7 Decellularization”: I believe that this section would benefit form a couple of additional references. For instance, Aulino et al. (Int J Med Sci 2015 PMID: 25897295) and Teodori et al. (Front Physiol 2014 PMID: 24982637) reporting on the plasticity of muscle-derived scaffolds or something similar with other decellularized organs.

5) Lines 734 etc. “9.6 Cardiac Tissue Engineering”: This section is inadequate in respect to the current amount of research in the field and should be extended. For instance, there are interesting papers with chitosan-based patches, such as Domenge et al (Acta Biomater 2021 PMID: 33161185) and Fiamingo et al (Biomacromolecules 2016 PMID: 27064341), but many other examples are available in the literature.

MINOR REMARKS

1) line 110. What does “red fibrillar” mean? Is “red” a typo?

2) I suggest replacing reference 33 with Perniconi B et al. 2011 in Biomaterials (PMID: 21802724): the latter is an original research article demonstrating exactly the point in the text, rather than a review article.

3) line 167, “gru delivery “ shall become “drug delivery agents” or something similar. Line 174 “its” shall be “their”. Line 182 “in the modern world” (??)

4) line294, “thermal energy acts as a solvent to separate the phases”: is it sure that the sentence is correct on a physico-chemical point of view? Line 888 “Contains information on 888 medical studies in human volunteers3 shall be “It contains…”.

5) Typo in ref # 122 numbering: “122. 122. Agrawal…”

Round 2

Reviewer 2 Report

Dear authors,

I can not recommend this review for publication because the information suggested in Response to Reviewer 2 Comments is principal other than in the revised paper. 

We added two new segments and update information with new references such as section 4.8 - "Cell sheets", page 15 (line 463) and section 5 - "smart materials" (line 492).

Information in the revised paper is absent.

section 4.8 - "Cell sheets" is absent.

section 5 - "smart materials" is absent.

We have cited the references: 

https://doi.org/10.3390/polym14194245

10.1056/NEJMoa040455

https://doi.org/10.1021/la036139f

https://doi.org/10.1007/978-1-0716-2261-2_15         https://doi.org/10.1016/j.colsurfb.2014.03.049

These references are absent in the list of references.

Also, as the reviewer suggests, figure 1 has been modified to include the smart materials for tissue engineering in the timeline.

Fig. 1 was not modified.

I believe there is a misunderstanding here, and the version of the revised paper that is available for review is not the final one, but this has to be corrected.

Author Response

Please see attached cover letter

Round 3

Reviewer 2 Report

The review can be accepted in the present form.